# Evaluating the diurnal cycle of South Atlantic stratocumulus clouds as observed by MSG-SEVIRI

Chellappan Seethala[1], Jan Fokke Meirink[2], Ákos Horváth[3], Ralf Bennartz[4], Rob Roebeling[5]

[1]Finnish Meteorological Institute, Kuopio, Finland

[2]Royal Netherlands Meteorological Institute, De Bilt, Netherlands

[3]University of Hamburg, Hamburg, Germany

[4]Vanderbilt University, Nashville, TN & University of Wisconsin – Madison, Madison, WI.

[5]EUMETSAT, Darmstadt, Germany

*Correspondence to*: Chellappan Seethala (seethala.chellappan@fmi.fi)

**Abstract**

Marine stratocumulus (Sc) clouds play an essential role in the earth radiation budget. Here, we compare liquid water path (LWP), cloud optical thickness ($\tau$), and cloud droplet effective radius ($r_e$) retrievals from two years of collocated Spinning Enhanced Visible and InfraRed Imager (SEVIRI), MODerate resolution Imaging Spectroradiometer (MODIS), and Tropical Rainfall Measuring Mission Microwave Imager (TMI) observations, estimate the effect of biomass burning smoke on passive imager retrievals, as well as evaluate the diurnal cycle of

South Atlantic marine Sc clouds.

The effect of absorbing aerosols from biomass burning on the retrievals was investigated using aerosol index (AI) obtained from the Ozone Monitoring Instrument (OMI). SEVIRI and MODIS LWPs were found to decrease with increasing AI relative to TMI LWP, consistent with well-known negative visible/near-infrared retrieval biases in $\tau$ and $r_e$. In the aerosol-affected months of July-August-September, SEVIRI LWP – based on the 1.6-µm $r_e$ – was

biased low by 14 g m$^{-2}$ (~16 %) compared to TMI in overcast scenes, while MODIS LWP showed a smaller low bias of 4 g m$^{-2}$ (~5 %) for the 1.6-µm channel and a high bias of 8 g m$^{-2}$ (~10 %) for the 3.7-µm channel compared to TMI. Neglecting aerosol-affected pixels reduced the mean SEVIRI-TMI LWP bias considerably. On a two-year data base, SEVIRI LWP had a correlation with TMI and MODIS LWP of about 0.86 and 0.94, respectively, and biases of only 4–8 g m$^{-2}$ (5–10 %) for overcast cases.

The SEVIRI LWP diurnal cycle was in good overall agreement with TMI except in the aerosol-affected months. Both TMI and SEVIRI LWP decreased from morning to late afternoon, after which a slow increase was observed. Terra and Aqua MODIS mean LWPs also suggested a similar diurnal variation. The relative amplitude of the two-year mean and seasonal mean LWP diurnal cycle varied between 35–40 % from morning to late afternoon for overcast cases. The diurnal variation in SEVIRI LWP was mainly due to changes in $\tau$, while $r_e$ showed only little

diurnal variability.

**1. Introduction**

Changes in marine boundary layer (MBL) clouds over eastern subtropical oceans and associated differences in cloud radiative forcing are thought to be the main source of uncertainty in climate feedback simulations (Bony and

Dufresne, 2005; Meehl et al., 2007, Zelinka et al., 2017). Climate models do not yet adequately parameterize the physical and dynamical processes affecting the formation of these clouds and fail to represent their variability on different time scales. Thus, understanding MBL cloud variability and its driving mechanisms remains crucial.

Marine stratocumulus (Sc), the dominant cloud type prevalent over eastern subtropical oceans, plays a vital role in radiation budget calculations because it reflects most of the incoming solar radiation back to space while having little effect on terrestrial radiation. Marine Sc clouds tend to form over relatively cold sea surface temperatures (SSTs), within a shallow, well-mixed MBL capped by strong subsidence and a strong temperature inversion (e.g., Albrecht et al., 1995; Norris, 1998; Wood and Hartmann, 2006; Sandu et al., 2010). Several studies investigated the synoptic to inter-annual variability and the driving mechanisms of these clouds from both an observational and a modeling perspective (e.g., Klein and Hartmann, 1993; Klein et al., 1995; Bretherton and Wyant, 1997; Wood and Bretherton, 2006; Eastman et al., 2011; Wood, 2012; Painemal et al., 2012, 2013a, 2015; Adebiyi et al., 2015; Adebiyi and Zuidema, 2016; Horowitz et al., 2017; Kar et al., 2018; Lu et al., 2018).

Marine Sc clouds are prevalent throughout the year and exhibit an explicit diurnal cycle (Minnis and Harrison, 1984; Wood et al., 2002, 2012). The daily maximum in marine Sc clouds tends to occur during the early morning hours before sunrise, while the minimum usually occurs in the afternoon (Minnis et al., 1992; Rozendaal et al., 1995; Bretherton et al., 1995; Wood et al., 2002). During daytime, shortwave absorption by clouds effectively reduces or even cuts off the transport of heat and moisture from the surface into the cloud layer, resulting in a decoupled MBL (Nicholls, 1984; Betts, 1990); simultaneously, enhanced cloud-top entrainment of dry air from above promotes a weaker inversion (Duynkerke et al., 2004), which leads to thinner or even disappearing clouds. During the night, on the other hand, strong longwave radiative cooling near cloud top produces negative buoyancy and, hence, a vertically well-mixed stable MBL (James, 1957; Moeng et al., 1992; Bretherton and Wyant, 1997), which increases cloud amount. Previous studies documented that subtropical Sc plays a significant role in the entire tropical response to climate perturbations (Miller, 1997), and underestimating the amount of these clouds in global climate models can lead to a positive SST bias as large as ~5 K (Ma et al., 1996). GCMs often fail to capture the diurnal variation of important processes in the cloud-topped MBL, such as the reduction of cloud fraction and the likelihood of decoupling in the afternoon (Abel et al., 2010; Medeiros et al., 2012). Wilson and Mitchell (1986) and Rozendaal et al. (1995) also demonstrated that introducing, or simply altering the resolution of, the diurnal cycle of these clouds in a GCM could trigger cloud radiative forcing both at the surface and at the top-of-atmosphere. Moreover, in the Fourth Assessment Report of the Intergovernmental Panel on Climate Change Forster et al. (2007) highlighted the diurnal cycle of stratiform clouds as one of the major uncertainties in current estimates of cloud radiative forcing. Comparisons of observations with models also revealed large and potentially systematic errors in the modeled diurnal cycle (O'Dell et al., 2008; Roebeling and van Meijgaard, 2009; Greuell et al., 2011).

To fully evaluate the diurnal cycle of Sc clouds, reliable observations with high spatial and temporal resolution are needed; the paucity of such data is one of the main reasons for the current level of uncertainty. A few studies took advantage of measurements available from intensive field campaigns, satellites, and model simulations

to investigate the diurnal variations of these clouds. Notably, Blaskovic et al. (1990) evaluated the diurnal cycle of northeast Pacific Sc off the California coast using observations during the First International Satellite Cloud Climatology Project Regional Experiment (FIRE). Painemal et al. (2013a) reported that cloud top height and cloud fraction over the southeast Pacific were increased in the early morning hours and reached a minimum in the afternoon. Most recently, Painemal et al. (2017) evaluated the diurnal cycle of cloud entrainment rate over northeast

Pacific marine boundary layer clouds based on geostationary satellite retrievals and a mixed-layer model, and reported that the cloud top height tendency term dominates the entrainment. Ciesielski et al. (2001) evaluated the diurnal variation of northeast Atlantic Sc from the Atlantic Stratocumulus Transition Experiment (ASTEX). Zuidema and Hartmann (1995) and Wood et al. (2002) studied the diurnal variation in liquid water path (LWP) based on observations from microwave imagers. Rahn and Garreaud (2010) and Burleyson et al. (2013) analyzed the

diurnal cycle of southeast Pacific Sc using the Variability of the American Monsoon Systems' Ocean-Cloud-Atmosphere-Land Study Regional Experiment (VOCALS-REx) datasets. Kniffka et al. (2014) studied the temporal and spatial characteristics of LWP of different types of clouds from SEVIRI data, for most of Europe and Africa. In general, all of these studies revealed an early morning maximum and afternoon minimum in cloud amount and LWP, linked to solar insolation/absorption. Rozendaal et al. (1995) and Wood et al. (2002) showed that the

amplitude of diurnal variations in cloud amount and LWP could exceed 20 % of the mean value. These studies, however, did not consider diurnal variations in cloud optical thickness ($\tau$) or cloud droplet effective radius ($r_e$) and were usually based on limited measurements from a single instrument, the uncertainties of which were not well characterized. In contrast, Painemal et al. (2012) evaluated the diurnal cycle of LWP, $\tau$, and $r_e$ for southeast Pacific Sc based on GOES-10 (Geostationary Operational Environmental Satellite-10) visible/near-infrared and microwave

satellite observations, as well as in-situ cloud probe data but only for a two-month period. They noted that variations in $\tau$ drive the diurnal cycle of LWP mostly.

In this study, we investigate the diurnal variations of southeast Atlantic Sc clouds. This geographic domain is notable for its unique feature that part of the year a smoke layer transported from the African continent resides above the Sc deck, which poses a challenge to the retrieval of aerosol and cloud properties from space. In recent

years several field campaigns have been initiated, to investigate aerosol-cloud interactions and their role in climate,

some of them in our region of interest (Zuidema et al., 2016). Satellite observations of cloud properties in this region are provided by the geostationary Meteosat Second Generation (MSG) Spinning Enhanced Visible and InfraRed Imager (SEVIRI) CLoud property dAtAset using SEVIRI - Edition 2 (CLAAS-2) from the Satellite Application Facility on Climate Monitoring (CM SAF) (Benas et al., 2017). The purpose of our study is to evaluate the CLAAS-2 cloud properties using Version 7.1 TMI (Tropical Rainfall Measuring Mission (TRMM) Microwave Imager) (Wentz 2018) and Collection 6 MODIS (Moderate Resolution Imaging Spectro-radiometer) retrievals (Platnick et al., 2017). In this process, the effect of above-cloud aerosols on LWP retrievals from the SEVIRI and MODIS passive imagers is quantified, and particular attention is paid to the diurnal cycle of the Sc clouds in the South Atlantic, which is a somewhat neglected region as most previous studies focused on the North or South Pacific (west of California and Chile). The main strength of our study is the use of an extensive two-year dataset, which allows us to investigate the seasonal variation of the diurnal cycle. SEVIRI's higher temporal resolution of 15 minutes allows examining the diurnal cycle with greater detail than offered by earlier GOES instruments. We only consider non-raining warm liquid clouds to minimize significant retrieval uncertainties associated with the presence of rain and ice clouds at higher altitudes. Retrieval artifacts related to absorbing aerosols (e.g., Haywood et al., 2004) have been evaluated and aerosol-affected grid boxes have subsequently been removed from the analysis.

The paper is structured as follows. A description of our datasets including retrieval artifacts and uncertainties is provided in Section 2. The comparison methodology is described in Section 3. Section 4 discusses retrieval biases related to the presence of smoke from continental biomass burning over clouds and analyzes spatial distributions, comparison statistics, and diurnal variations of Sc properties from SEVIRI, TMI, and Terra and Aqua MODIS on seasonal and two-year timescales. Finally, a summary is offered in Section 5.

## 2. Satellite datasets

### 2.1 Visible/Near-infrared (VIS/NIR) retrievals

#### 2.1.1 Spinning Enhanced Visible and InfraRed Imager (SEVIRI)

SEVIRI is an optical radiometer onboard the MSG geostationary satellite series operated by the European Organization for the Exploitation of Meteorological Satellites (EUMETSAT). SEVIRI measures radiances in 12 spectral bands including 4 visible/near-infrared (VIS/NIR) channels (0.6–1.6 μm plus a broadband high-resolution VIS channel) and 8 IR channels (3.9–13.4 μm). It has a spatial resolution of 3x3 km$^2$ at nadir and a repeat frequency of 15 minutes for full-disk images covering Europe, Africa, and the Atlantic Ocean.

The CM SAF CLAAS-2 climate data record is described in Benas et al. (2017). Part of the cloud processing software is the CPP (cloud physical properties) algorithm, which retrieves cloud optical thickness and cloud droplet effective radius based on measured reflectances in the 0.63-µm and 1.6-µm channels. The retrieval scheme is based on earlier bispectral methods (hereafter also referred to as visible/near-infrared or VIS/NIR technique) that retrieve cloud optical thickness and cloud droplet effective radius from satellite radiances at

wavelengths in the (for clouds) non-absorbing visible and the moderately absorbing solar infrared part of the spectrum (Nakajima and King 1990; Han et al. 1994; Nakajima and Nakajima, 1995; Watts et al., 1998; Roebeling et al., 2006). The liquid water path is computed from the retrieved cloud optical thickness ($\tau$) and cloud droplet effective radius ($r_e$) as

$$\mathrm{LWP} = \frac{2}{3}\,\tau r_{e(1.6\mu m)}\rho_l, \text{ where } \rho_l \text{ is the density of liquid water (Stephens 1978).} \tag{1}$$

The SEVIRI retrievals are available only during daytime and are performed assuming plane parallel, vertically homogeneous clouds. Because $r_e$ is not well constrained by the measured 1.6-µm channel reflectance for thin clouds, it is weighted towards a climatological a priori value of 8 µm for pixels with $\tau \leq 4$ -similar to the handling of small optical thicknesses in optimal estimation methods. The relationship used to weight the $r_e$ retrieval is,

$$r_{e,assign} = \ r_{e,clim}\,(1 - w) + \ r_{e,ret}\,w \tag{2}$$

where, $w = \ 1/(1 + e^{(-1.25(\tau_{ret} - \tau_{w,clim}))})$; $r_{e,clim} = 8$ µm; $\tau_{w,clim} = 2.5$

In part of our analysis, a $\tau > 3$ threshold is applied to minimize the impact of strongly weighted effective radii for thin clouds on the results. The SEVIRI shortwave channels were calibrated with Aqua-MODIS as described in Meirink et al. (2013). More details on the CPP retrieval algorithm are provided in CM SAF (2016).

**2.1.2 Moderate Resolution Imaging Spectroradiometer (MODIS)**

MODIS is the flagship instrument aboard the Terra and Aqua polar orbiter satellites. Terra has a 10:30am descending node sun-synchronous orbit, while Aqua has a 13:30pm ascending node sun-synchronous orbit. Terra and Aqua MODIS image the entire Earth's surface every 1 to 2 days, acquiring data in 36 spectral bands. The MODIS Collection 6 (C6) cloud property datasets (Platnick et al., 2017) with 1x1 km$^2$ spatial resolution from both

Terra (MOD06) and Aqua (MYD06) have been used in this study.

       Similar to CLAAS-2 SEVIRI, the MODIS C6 algorithm uses the VIS/NIR technique to retrieve cloud properties. Over ocean, the 0.86-µm band is used for optical thickness information in conjunction with one of three water-absorbing near-infrared bands located at 1.6, 2.2, and 3.7-µm, which are particularly sensitive to droplet

effective radius. Although all three near-infrared channels generally observe the upper portion of clouds, the vertical sampling of droplets becomes progressively deeper from 3.7 to 1.6-μm due to decreasing absorption (Platnick, 2000).

The C6 algorithm is a revamped version of the Collection 5 (C5) algorithm that has gone through several updates to improve performance. Modifications include improved radiative transfer and lookup tables with finer $\tau$ and $r_e$ bins, redesigned cloud thermodynamic phase detection based on a variety of independent tests, and separate spectral retrievals of $\tau$, $r_e$, and derived LWP for channel combinations using the 1.6, 2.2, and 3.7-μm bands. Differences in $r_e$ between C5 and C6 are evaluated in Rausch et al. (2017). Depending on a subpixel heterogeneity index, the properties of partly cloudy pixels are listed separately and the algorithm also provides retrieval failure metrics for pixels where the observed reflectances fall outside the LUT solution space.

**2.1.3 Known uncertainties in VIS/NIR retrievals**

While these datasets offer excellent resources for investigating warm, overcast single-layer clouds, they are subject to certain retrieval artifacts due to algorithm assumptions and complexities in the retrieval technique. The VIS/NIR cloud property retrievals rely on 1-D radiative transfer model-generated LUTs, which do not account for subpixel cloud heterogeneity and 3-D cloud structure, and that could lead to significant biases in retrieved cloud properties for inhomogeneous and partially cloudy scenes. Cloud vertical stratification is essential to consider when computing LWP. Although MODIS retrieves effective radius at three separate water-absorbing channels, 1.6, 2.2, and 3.7-μm, all three are most sensitive to near cloud-top properties (Platnick 2000; Zhang and Platnick 2011). Hence, the LWP derived by combining retrieved $\tau$ and retrieved $r_e$ from any one of the near IR channels could potentially under- or overestimate the true value depending upon the actual cloud stratification. For stratocumulus that typically follows a sub-adiabatic $r_e$ profile, bigger droplets will be located near cloud top, and thus the derived LWP could be an overestimate. As a first-order correction, an adiabatic model is proposed by Wood and Hartmann (2006), which results in a ~17 % reduction from the standard vertically homogeneous LWP in eq. 1 (Bennartz 2007; Bennartz and Rausch, 2017). More details about the retrieval uncertainties of the VIS/NIR technique can be found in Horváth and Davies (2007), Seethala and Horváth (2010), Horváth et al. (2014), Zhang et al., (2012), Grosvenor et al., (2018) and references therein.

**2.2 TRMM Microwave Imager (TMI)**

TMI was a 5-channel, dual-polarized, passive microwave imager onboard the Tropical Rainfall Measuring Mission (TRMM) satellite that was operational between December 1997 – April 2015, continuously monitoring the tropics between 40° S and 40° N. Unlike the sun-synchronous polar orbiters hosting the similar SSM/I (Special Sensor Microwave/Imager) instruments, the TRMM satellite precessed west to east in a semi-equatorial orbit, producing data at different local times. The radiometer measured microwave radiation at 10.7, 19.4, 21.3, 37, and 85.5 GHz. The Wentz absorption-emission based algorithm (Wentz, 1997; Wentz and Spencer, 2000; Hilburn and Wentz, 2008) is used to retrieve meteorological parameters such as sea surface temperature (SST), surface wind speed (W), water vapor path (V), liquid water path (LWP), and rain rate (R) over the ocean. Our primary interest, LWP, is mainly derived from 37-GHz observations at a native resolution of 13 km, although here we used the 0.25° gridded product available from Remote Sensing Systems (RSS). The error characteristics of TMI data are similar to those of the RSS SSM/I and AMSR-E (Advanced Microwave Scanning Radiometer for Earth Observing System) products, as all microwave retrievals are produced by the same unified algorithm. Various sources of potential errors are documented in Horváth and Gentemann (2007), O'Dell et al. (2008), Seethala and Horváth (2010), Elsaesser et al. (2017), and Greenwald et al. (2018). Because the diurnal cycle is targeted here, the non-sun-synchronous TMI observations are particularly useful. The precessing orbit of TRMM allows for a comparison of observations at different local times, which cover the entire diurnal cycle over the course of a month.

TMI data were recently reprocessed using the significantly improved version 7.1 (V7.1) of the radiometer data processing algorithm (Wentz, 2015). The following major modifications were introduced: TMI brightness temperatures were recalibrated using the same procedures applied to all other RSS microwave products, the previously removed small negative LWP values are now reported, some large geolocation errors were corrected, the roll of the satellite was recalculated, the radiation contribution from the emissive antenna itself was removed, and Radio Frequency Interference (RFI) from the cold mirror was minimized. This improved V7.1 TMI product available at www.remss.com was utilized in this study.

**2.3 Ozone Monitoring Instrument (OMI)**

Areas affected by biomass burning smoke or desert dust were identified using OMI ultraviolet Aerosol Index (AI). OMI AI represents the deviation of measured 354-nm radiance from model estimates calculated for a purely molecular atmosphere bounded by a Lambertian surface, with positive values indicating the presence of absorbing aerosols (Torres et al. 2007). A distinguishing feature of OMI AI is its ability to detect absorbing aerosols above

(and even mixed with) clouds. Specifically, we used the 0.25° resolution daily Level-2 gridded product (OMAERUVG).

**3. Comparison methodology**

For our study we used two years of data (December 2010 – November 2012) from SEVIRI, TMI, and Terra- and Aqua-MODIS. We consider JJA (June-July-August), SON (September-October-November), DJF (December-January-February), and MAM (March-April-May) respectively to represent austral winter, spring, summer, and autumn; henceforth, 'seasonal' refers to an average over a given season in 2 consecutive years. SEVIRI pixel-level data were averaged down to TMI's $0.25^{o}$ x $0.25^{o}$ resolution, only using SEVIRI retrievals within ±7.5 minutes of the TMI observation time. Note that SEVIRI and MODIS LWPs are representative of the in-cloud LWP. For compatibility with the TMI gridbox-mean LWP, we multiplied SEVIRI LWP with the successful cloud retrieval fraction (henceforth referred to as "liquid cloud fraction" or LCF) within the TMI gridbox. Similarly, when matching $\tau$, $r_e$, and LWP from SEVIRI and MODIS, both datasets were averaged down to $0.25^{o}$ x $0.25^{o}$ resolution, using the same temporal collocation criterion of ±7.5 minutes, and the MODIS LWP was also scaled by the corresponding LCF. In order to minimize the impact of SEVIRI $r_e$ values that were strongly weighted towards a climatological a priori value in thin clouds, only those $0.25^{o}$ x $0.25^{o}$ gridboxes (and the corresponding TMI, SEVIRI, and MODIS retrievals) were included in the analysis of overcast scenes, which had a gridbox-mean SEVIRI $\tau > 3$.

Our study domain is a $70^{o}$ x $40^{o}$ ($50^{o}$ W-$20^{o}$ E, $35^{o}$ S-$5^{o}$ N) area in the Southeast Atlantic. Over the relatively cold SSTs near the Namibian coast extensive sheets of marine Sc clouds form, which transition into scattered trade Cu as they are advected towards warmer ocean near the equator. Decks of subtropical marine Sc, scattered Cu, and occasionally deep convective clouds cover the study domain. We however restricted our analysis to marine Sc clouds.

Because microwave and optical techniques represent fully independent approaches, each having their own shortcomings, the analysis of retrieval discrepancies does not necessarily establish absolute accuracies. A considerable number of studies have investigated the differences between LWP retrievals based on passive microwave and VIS/NIR satellite observations (Bennartz, 2007; Borg and Bennartz, 2007; Horváth and Davies, 2007; Horváth and Gentemann, 2007; Wilcox et al., 2009; Greenwald, 2009; Seethala and Horváth, 2010; Horváth et al., 2014, Cho et al., 2015; Greenwald et al., 2018). The major shortcomings of microwave retrievals were found to be the uncertain retrieval of LWP in the presence of rain and a wet (positive) bias of 10-15 g m$^{-2}$ in broken cloud

fields. However, V7.1 TMI data now includes the small negative LWP values that were previously discarded in V4 data, leading to a significantly reduced microwave wet bias (Greenwald et al., 2018).

The major issues affecting VIS/NIR measurements are the dependence of retrievals on cloud fraction, variations with sun-view geometry, horizontal and vertical subpixel inhomogeneity, 3D radiative effects, and the presence of aerosols/cirrus above the liquid cloud layer. Agreement between the VIS/NIR and microwave techniques is generally better for more stratiform clouds, where a near-adiabatic cloud liquid water profile can be assumed. To minimize these retrieval problems, we examine the diurnal characteristics of only low-level non-raining warm (liquid) clouds, which typically dominate the South Atlantic marine Sc domain. Additional criteria are applied to reduce as much as possible the influence of rain and ice clouds: gridboxes are included only if flagged as confident liquid clouds with valid LWP retrieval, cloud top temperature (CTT) > 275 K, ice fraction = 0 in SEVIRI and MODIS retrievals, and rain rate = 0 in TMI retrievals. Our study domain is also affected by continental biomass burning during austral winter and spring, which in turn affects VIS/NIR cloud retrievals; therefore, special attention is paid to the analysis of retrieval artifacts related to the presence of smoke over the Sc deck.

We noticed that the extent and location of South Atlantic Sc clouds vary from month to month; hence we opted to define the Sc domain dynamically, rather than selecting a fixed rectangular area as study domain. Thresholding the spatial mean map of liquid cloud fraction (LCF) and the heterogeneity parameter ($H_\sigma$ = SEVIRI 0.63-µm reflectance standard deviation / mean reflectance) was found to delineate Sc regions in good agreement with visual observations. To precisely define the Sc domain, we used a region-growing algorithm to find adjacent, connected grid-boxes with LCF > 80 %. The identified Sc regions were typically within $20^{o}$ W – $20^{o}$ E and $5^{o}$ S – $35^{o}$ S. Cloud properties were separately evaluated for two cases:

1. 'all-sky': including all grid boxes from the identified Sc domain.

2. 'overcast': only including grid boxes from the identified Sc domain, which had an LCF ≥ 95 % and a mean SEVIRI τ > 3. These criteria were imposed to minimize retrieval artifacts related to broken clouds as well as thin clouds for which the SEVIRI $r_e$ retrieval in particular is relatively uncertain.

**4. Results**

**4.1 Effect of biomass burning smoke on SEVIRI and MODIS retrievals**

This section presents the analysis of the effect of smoke and/or aerosols above marine Sc on passive VIS/NIR imager retrievals of cloud properties. Our study domain, especially the Sc region located off the Namibia coast, is

severely influenced by biomass burning on the African continent, as it produces episodic plumes of dark smoke that drift over the southeast Atlantic Ocean during the dry season JJASO (June-through-October). Beneath the elevated smoke layer, there is a persistent deck of bright marine Sc clouds. Previous research (Hobbs, 2002; McGill et al., 2003; Wilcox, 2010) has shown that the smoke is typically located in layers (at 2 to 4 km altitude) that are vertically separated from the Sc clouds below (at ~1.5 km altitude) and, hence, direct microphysical interaction between the aerosols and the Sc is often inhibited by the strong temperature inversion above the cloud layer. However, more recent studies e.g., Rajapakshe et al. (2017) reported that smoke layers are closer to the cloud layer, and significantly enhance the brightness of stratocumulus over there (Lu et al. 2018). Recently, several studies evaluated the dynamical and climatological impacts of the presence of smoke above Sc clouds from both modeling as well as satellite and/or field campaign measurements (Adebiyi et al., 2015; Adebiyi and Zuidema, 2016; Zuidema et al., 2016; Das et al., 2017; Horowitz et al., 2017; Chang and Christopher, 2017; Lu et al., 2018; Kar et al., 2018). When smoke resides above low-level clouds, the observed visible channel (0.6- or 0.8-μm) reflectance is reduced due to absorption by smoke, which is not taken into account in the LUTs and can introduce a negative bias in the retrieved $\tau$ as well as $r_e$, and hence in LWP. According to Haywood et al. (2004), this negative bias in the 1.6-μm $r_e$ is significantly larger than that in the 2.1-μm $r_e$ (which is estimated to be less than 1 μm), while the bias in retrieved $\tau$ can be up to 30 %. Previous studies also noticed a domain-mean underestimation of ~3 to 6 g m$^{-2}$ in MODIS LWP over the South Atlantic Sc region in the presence of absorbing aerosols (Bennartz and Harshvardhan, 2007; Wilcox et al., 2009; Seethala and Horváth, 2010). Therefore, we need to quantify the impact of absorbing aerosols on SEVIRI and MODIS VIS/NIR retrievals in our Sc domain for our study period. The presence of absorbing aerosols can be diagnosed using the OMI Aerosol Index (AI), because large positive AIs correspond to absorbing aerosols, such as dust and smoke, and small positive or negative AIs correspond to non-absorbing aerosols and clouds.

Figure 1a depicts the spatial distribution of average OMI aerosol index during JAS for 2011 and 2012, with the black contour representing the Sc region. It is clear that absorption by smoke is highest near the Namibian coast and decreases away from shore. The locations of greater cloud amount partly coincide with the locations of larger AIs. The spatial distribution of SEVIRI and TMI LWP and their bias for overcast conditions are shown in Figs. 1b-d. Near the coast where the smoke absorption is stronger, SEVIRI LWPs increasingly underestimated the TMI LWPs (SEVIRI values were approximately half of the corresponding TMI values). Over the smoke-free areas of the stratocumulus region, on the other hand, SEVIRI-retrieved LWPs were slightly higher than TMI LWPs. The

domain-mean TMI LWP is 85 g m$^{-2}$, whereas the mean SEVIRI LWP is only 71 g m$^{-2}$, indicating an LWP low bias

of 14 g m$^{-2}$ or ~16 % in SEVIRI retrievals.

In Fig. 2, cloud properties from TMI, SEVIRI, and MODIS retrievals are binned into AI bins of 0.5 for the

overcast Sc conditions. In the SEVIRI 1.6-µm $r_e$ retrievals, a steady and strong decrease from 11 to 6 µm is

observed, while the $\tau$ decrease is weaker from 10.8 to 9 with AI increasing from 0 to 3.5. As a result, SEVIRI LWP

sharply decreases from 86 to 45 g m$^{-2}$ over the same AI range. TMI LWP, in contrast, increases from 84 to 101 g m$^{-2}$

between clean and increasingly polluted regions. For overcast grid boxes with little to no smoke absorption (AI <

0.5), SEVIRI LWP agrees well with TMI LWP, having only a 2 g m$^{-2}$ high bias. However, SEVIRI has a low bias of

6–25 g m$^{-2}$ for moderate AI between 1 and 2, and a large negative bias < -40 g m$^{-2}$ for grid-boxes with AI > 2.5; the

bias increases linearly with AI.

    Considering that cloud amount happens to be spatially correlated with AI and that microwave retrievals are

unaffected by absorbing aerosols, the increase in TMI LWP with increasing AI, that is closer to shore, seems

plausible. Because the variability of LWP is mostly controlled by $\tau$ rather than $r_e$ in the absence of smoke-induced

retrieval biases (Seethala and Horváth, 2010; Painemal et al., 2012), the microwave retrievals suggest that the true $\tau$

should also increase with AI. Taken together, the microwave and VIS/NIR retrievals imply that SEVIRI $\tau$ is

increasingly underestimated as AI increases, in line with Haywood et al. (2004). The low bias in SEVIRI LWP in

smoke-affected areas arises from the combination of the negative $\tau$ and $r_e$ retrieval biases. A similar underestimation

is reported in aircraft retrievals of $\tau$ and $r_e$ for a stratus deck residing below an absorbing aerosol layer (Coddington

et al., 2010).

    Interestingly, a systematic overall increase in LWP with AI as indicated by TMI LWP in Fig. 2a was also

noticed in previous observational and modeling studies, e.g., Johnson et al. (2004), Wilcox (2010), Randles and

Ramaswamy (2010), Adebiyi et al. (2015), Adebiyi and Zuidema (2016). While this could partly be explained by the

fortuitous spatial correlation between higher aerosol loads and thicker clouds in this Sc region, these studies argue

that strong atmospheric absorption by the smoke warms the 700 hPa air temperature and increases upward motion.

This increased buoyancy inhibits cloud-top entrainment and promotes a stronger inversion, thereby helping to

preserve humidity and cloud cover in the MBL, resulting in increased cloud amount and LWP compared to a smoke-

free environment. Similar to our SEVIRI results, Bennartz and Harshvardhan (2007), Wilcox et al. (2009), and

Seethala and Horváth (2010) also noted a systematic MODIS LWP underestimation in Sc off southern Africa during

the biomass burning seasons. Painemal et al. (2014) also noted a decrease in MODIS $r_e$ despite increased LWP north of 5° S during the biomass burning season.

Retrieval discrepancies due to the presence of absorbing aerosols above Sc clouds were also evaluated between SEVIRI and MODIS. Unlike the TMI microwave technique, SEVIRI and MODIS rely on VIS/NIR channels for cloud property retrieval and hence are heavily impacted by above-cloud aerosols. SEVIRI uses 0.63-µm reflectances, whereas MODIS uses 0.86-µm reflectances over ocean, primarily to acquire $\tau$. According to the radiative transfer calculations of Haywood et al. (2004), aerosols over bright Sc reduce both the 0.63- and the 0.86-µm radiances, but the reduction is more pronounced in the former due to the wavelength dependence of aerosol optical thickness. Their calculations indicated a low bias of 2 and 6 in $\tau$ retrieved from 0.86-µm reflectances for a true $\tau$ of 10 and 20 respectively, and this low bias would be larger in $\tau$ retrieved from 0.63-µm reflectances. The water-absorbing channels used primarily to retrieve $r_e$ are 1.6-, 2.1-, and 3.7-µm for MODIS, whereas for CLAAS-2 only the SEVIRI 1.6-µm channel is used. The 3.7-µm and to a smaller extent the 2.1-µm retrieved $r_e$ are less affected by aerosols above clouds, because the constant $r_e$ lines are nearly parallel to the visible reflectance axis in the bispectral LUT. In contrast, the 1.6-µm based constant $r_e$ lines are less parallel to the 0.63- or 0.86-µm reflectance axis in the LUT and hence there is a stronger underestimation of 1.6-µm $r_e$. For example, according to Haywood et al. (2004), the 0.86/1.63-µm radiance pair produced a significant low bias in $r_e$ of about 3 µm for a cloud with actual $r_e$ of 10 µm, due to the apparent indirect effect induced by the decreased 0.86-µm radiance on non-parallel constant $r_e$ lines. This low bias will be even larger for the 0.63/1.63 radiance pair as used for the SEVIRI retrievals. As a result, SEVIRI $\tau$ and $r_e$ in smoke-affected regions are both expected to be smaller than their MODIS counterparts retrieved from the 0.86/1.63-µm radiance pair. In general, such absorbing-aerosol biases are more pronounced for bright optically thick clouds.

Figure 3 depicts the spatial distribution of SEVIRI-MODIS LWP, $\tau$, and $r_e$ differences for all three water absorbing MODIS channels, averaged for JAS 2011 and JAS 2012 overcast conditions. Within the Sc regime, SEVIRI $\tau$ is biased low by 1 compared to MODIS where the smoke absorption is highest. As expected, little variation is observed in the SEVIRI-MODIS $\tau$ bias as a function of the MODIS water-absorbing channel, because $\tau$ is mostly determined by the VIS channel reflectance. However, $r_e$ from the three MODIS channels and thus the corresponding SEVIRI-MODIS $r_e$ bias, dramatically differs over the largest smoke absorption areas. As discussed above, MODIS $r_e$ retrieved from the 3.7-µm channel is expected to be the least affected by absorption effects. As shown, MODIS 3.7-µm $r_e$ values are 2 to 5 µm larger than the SEVIRI 1.6-µm $r_e$, with the largest differences

occurring in grid boxes with the strongest smoke absorption effects (in the 1.6-μm channel). The low bias between SEVIRI $r_e$ and MODIS 2.1-μm $r_e$ is 1 to 3 μm, whereas it is only ~1 μm compared to MODIS 1.6-μm $r_e$, consistently over the Sc regime. As a result of the SEVIRI τ and $r_e$ low biases, the SEVIRI-MODIS LWP bias also increases from the MODIS 1.6-μm to the 3.7-μm channel. Not surprisingly, SEVIRI LWP agrees best with MODIS 1.6-μm LWP, with a typical bias of ±5 g m$^{-2}$ and a maximum bias of ~10 g m$^{-2}$ over areas with the strongest smoke

absorption. Compared to the MODIS 2.1- and 3.7-μm retrievals, the SEVIRI LWP bias ranges from 10-20 and 10-30 g m$^{-2}$, respectively, again showing the maximum over the strongest smoke absorption regions. These results are fully consistent with the differential absorption effects found by Haywood et al. (2004) and confirm that the 3.7-μm channel is the least affected by biomass smoke and generally performs best in aerosol-above-cloud situations.

Frequency histograms of SEVIRI minus MODIS LWP, τ, and $r_e$ biases, as well as the biases relative to

390 MODIS CPP for overcast conditions aggregated for JAS 2011 and JAS 2012 are shown in supplemental Fig. S1. SEVIRI τ is biased low by ~1 compared to MODIS. Compared to the 1.6-μm MODIS $r_e$, ~70 % of SEVIRI $r_e$ have a mean bias of -1.5 μm. Although SEVIRI $r_e$ are biased low compared to all three MODIS $r_e$ retrievals, the ~1 μm additional negative bias relative to the 2.1- and 3.7-μm $r_e$ indicates much smaller smoke-induced retrieval artifacts in these two channels. In general, the $r_e$ retrievals from SEVIRI tend to be lower than corresponding retrievals from the

395 three MODIS channels, with SEVIRI having about 1.5 μm to 2.5 μm lower $r_e$ values. The SEVIRI minus MODIS LWP distributions peak at about -10 g m$^{-2}$ irrespective of the MODIS channel used for the retrieval.

The mean MODIS LWPs are 80 g m$^{-2}$, 87 g m$^{-2}$, and 90 g m$^{-2}$ respectively for 1.6-, 2.1-, and 3.7-μm channel retrievals, while the corresponding mean SEVIRI LWP is 71 g m$^{-2}$. As shown in Fig. 2a, MODIS 1.6-μm retrieved LWP undergoes the largest decrease from 92 to 72 g m$^{-2}$ with AI. In clean cases, MODIS 1.6-μm LWP is

400 10 % higher than the SEVIRI 1.6-μm LWP. The difference between MODIS and SEVIRI LWP is even larger for the 2.1- and 3.7-μm channel retrievals due to the wavelength-dependent absorption effects.

A decrease from 12 to 9 μm is observed in MODIS 1.6-μm $r_e$, whereas both the MODIS 2.1- and 3.7-μm $r_e$ show a smaller decrease of ~1.5 μm with increasing AI (Fig. 2e). This, again, indicates the reduced effect of absorbing aerosols on 2.1- and 3.7-μm reflectances. Surprisingly, SEVIRI 1.6-μm $r_e$ are about 1.5 μm (2.5 μm and 3

405 μm) lower than MODIS 1.6-μm (2.1- and 3.7-μm channels) $r_e$, even in less polluted (AI < 0.5) overcast conditions.

MODIS τ decreased slightly until AI < 1.5, increased steeply until AI = 2.5, and then leveled-off in all three channels. However, SEVIRI and MODIS τ differ by 1 with MODIS being higher even in grid-boxes least affected by smoke. Taken together both τ and $r_e$ variations, MODIS LWPs show a decreasing trend with AI in all three

channels, with the largest decrease of ~20 g m$^{-2}$ seen in the 1.6-µm retrieval. The 2.1- and 3.7-µm MODIS LWPs show a reduction of only ~10 g m$^{-2}$. The SEVIRI minus MODIS differences in LWP, $\tau$, and $r_e$ increased with AI even in the common 1.6-µm channel; although differences were the smallest in this channel, especially for $r_e$. This is somewhat surprising, considering that the CLAAS-2 SEVIRI and MODIS C6 $\tau$-$r_e$ retrieval algorithms are rather similar, the SEVIRI 1.6-µm channel has been calibrated with the corresponding MODIS channel, and the comparison is done for the most favorable overcast condition. The finding that AI has a stronger impact on SEVIRI 1.6-µm LWP than on MODIS 1.6-µm LWP could be explained, as discussed above, by spectral differences in the visible channel used: for SEVIRI retrievals the 0.63-µm channel is used as a non-absorbing channel in contrast to the 0.86-µm channel for MODIS, the latter of which is less affected by aerosol absorption.

Because the presence of absorbing aerosols above Sc clouds introduces a large negative bias in both SEVIRI and MODIS $\tau$ and $r_e$ retrievals, in the remainder of this work we will exclude grid-boxes with AI > 0.1.

**4.2 Spatial distribution and mean statistics of SEVIRI, MODIS, and TMI cloud properties**

This section presents the results of the comparison of SEVIRI, MODIS, and TMI LWP retrievals, as well as the comparison of SEVIRI and MODIS $\tau$ and $r_e$ retrievals. Significant variation in the distribution and amount of clouds is observed over the Sc region from month to month. During SON, we observe frequent Sc clouds with large spatial extent. During JJA there are relatively fewer clouds that are shifted slightly to the north. The lowest cloud fractions are seen during DJF and MAM. From a surface-based cloud climatology, Klein and Hartmann (1993) also showed that there is strong seasonal variability in the amount of Sc clouds, which is closely tied to the seasonal cycle of static stability. Over the South Atlantic Sc region, SON had the largest lower tropospheric stability (LTS), and DJF had the smallest. The strongest net cloud radiative effect also occurred during August through November, which further motivates us to examine the seasonal variability of these clouds.

The spatial distributions of two-year-mean SEVIRI cloud properties and TMI LWP for the overcast condition are shown in Fig. 4, whereas the results for the all-sky case are shown in Fig. S2. In the all-sky case, the spatial distribution of LWP indicates that over the marine Sc region the measurement techniques show good agreement, but SEVIRI overestimates TMI by ~15 g m$^{-2}$ in smooth coastal fog. In contrast, the two-year mean SEVIRI LWP is much lower than the corresponding TMI mean LWP in regions with generally lower cloud fractions and clouds with structured tops. This could be either due to a high bias in TMI LWP in broken scenes (Seethala and Horváth, 2010; Greenwald et. al., 2018) or an enhanced plane-parallel bias in broken more heterogeneous scenes

underestimating $\tau$ and overestimating $r_e$ in SEVIRI 3 km retrievals. In the Sc region, SEVIRI $\tau$ varies from 6 to 11 and $r_e$ ranges between 8 and 14 µm. The two-year-mean liquid cloud fraction varies between 75 % and 100 %. The mean statistics also show robust skill in LWP retrieval for both SEVIRI and TMI with a high correlation of 0.89 for the Sc regime. Both TMI and SEVIRI show a mean LWP of ~53 g m$^{-2}$ with negligible bias and a standard deviation of 24 g m$^{-2}$ for the study period.

In the overcast case over the Sc regime, the two-year mean LWP increases to 84 g m$^{-2}$ and 80 g m$^{-2}$ respectively for SEVIRI and TMI, i.e., the mean SEVIRI LWP is about 5 % larger than the mean TMI LWP. In this case, applying an adiabatic correction to SEVIRI LWP would lead to a larger bias of -10 g m$^{-2}$ (-12 %) and standard deviation of 28 g m$^{-2}$. The unbiased LWPs observed in the all-sky Sc case could be associated with the cancellation of errors between fully overcast and lower LCF grid-boxes within the domain. A higher mean $\tau$ of ~11 characterizes the overcast Sc case, whereas the mean $\tau$ is only about 7 in the all-sky case, suggesting the presence of optically thin clouds which are more prone to retrieval biases. Figure 5 shows a density scatterplot of TMI and SEVIRI LWPs in the overcast Sc region. Most data points are close to the one-to-one line, although at the lower end TMI LWP is slightly higher, while the reverse is true at the higher end -the same feature is also found in monthly and seasonal results.

The daytime-averaged two-year and seasonal statistics of SEVIRI and TMI LWP are listed in Table 1. Seasonally, in the overcast Sc domain, the average LWP varies from 73 to 92 g m$^{-2}$ in standard SEVIRI, 61 to 76 g m$^{-2}$ in adiabatic SEVIRI, and 73 to 82 g m$^{-2}$ in TMI retrievals. In the aerosol-free seasons of DJF and MAM, standard SEVIRI overestimates TMI LWP; applying an adiabatic correction to SEVIRI in these months brings the LWP bias within 5 %, similar to previous studies. The standard SEVIRI likely overestimates the actual LWP in the overcast Sc regime due to the overestimation of $r_e$, as the observed $r_e$ in the 1.6-µm channel corresponds to the top layer and is higher than the cloud layer-mean in sub-adiabatic stratocumulus. However, for JJA, when all three months were heavily affected by smoke aerosol, the standard SEVIRI already shows ~10 % lower LWP than TMI; therefore, applying the adiabatic correction would only enhance this negative bias. For SON, only September was heavily affected by aerosol for the analysis years we considered. As a result, the mean standard SEVIRI LWP was ~5 % larger than TMI LWP and applying adiabatic correction would lead to a ~14 % underestimation in SEVIRI LWP. We found that SEVIRI underestimates LWP more during the aerosol-affected months, even after excluding grid-boxes with AI > 0.1 Applying a stricter criteria by excluding grid-boxes with AI > 0 did not improve the results, hinting at residual OMI AI retrieval biases.

The spatial distribution of SEVIRI and MODIS cloud properties averaged for the study period for the overcast condition is shown in Fig. 6. In general, over the overcast Sc regime, MODIS retrieves higher LWPs in all three channels compared to SEVIRI, but outside the identified Sc regime, MODIS values are lower than SEVIRI

LWP with the exception that 1.6-µm (and to a certain extent 2.1-µm) MODIS LWPs are lower in the north, i.e., closer to the equator and higher in the south. This LWP bias pattern can be explained by the respective $r_e$ spatial variation shown in Fig. 6(f and i) and that in cloud heterogeneity, as the fractional cloud cover is greater than 95 % in this case. SEVIRI $r_e$ values are $1 - 2$ µm higher in the north closer to the equator, probably indicating the plane-parallel bias in the larger SEVIRI pixel. However, SEVIRI $r_e$ is $1 - 2$ µm lower in the south of the domain due to the

increased frequency of the climatological weighting of SEVIRI $r_e$ for lower $\tau$ values, while MODIS provides an actual retrieved $r_e$. MODIS 3.7-µm $r_e$ is consistently lower than SEVIRI likely because this channel is least affected by horizontal cloud heterogeneity and 3D cloud structure. The observed two-year-mean $\tau$ in the overcast Sc regime is 10.2 for SEVIRI, whereas it is 11.2 for MODIS, indicating SEVIRI mean $\tau$ is about 9 % lower than MODIS mean $\tau$. Similarly, the observed two-year-mean $r_e$ in the overcast Sc domain is 10.1 µm for SEVIRI, but for MODIS it

varies between $11.3 - 11.7$ µm depending on the absorption channel, indicating that SEVIRI mean $r_e$ is 11–12 % lower than MODIS $r_e$. The lower SEVIRI LWP value over the Sc regime is due to the combination of lower $\tau$ and lower $r_e$ values compared to MODIS. As expected, the $\tau$ bias varied little with MODIS absorption channel, whereas the bias in $r_e$ significantly depended on the MODIS NIR channel –MODIS 1.6-µm $r_e$ is consistently ~1 µm higher than the corresponding SEVIRI, however, MODIS 2.1- and 3.7-µm $r_e$ are 2 to 3 µm larger closer to the Namibia

coast, indicating the potentially still existing effect of smoke absorption in austral winter months in two-year mean regional distribution, even after discarding the pixels with OMI AI > 0.1.

The higher SEVIRI LWP values closer to the equator outside the identified Sc cloud regime are exclusively due to a 1–2 µm overestimation in $r_e$ compared to MODIS, as SEVIRI $\tau$ remain underestimated in almost all grid boxes in the study domain. The lower SEVIRI LWP values in the south of the domain outside the identified Sc

cloud regime are due to a combination of low biases in both SEVIRI $\tau$ (~1) and $r_e$ (~1 µm). This geographic variation in SEVIRI LWP could be caused by the combined contribution of two factors: (i) in thin clouds with $\tau < 4$, the SEVIRI CPP algorithm weighting $r_e$ with an a priori (climatological) value of 8 µm, but MODIS providing the retrieved values, and (ii) the plane-parallel bias in heterogeneous scenes causing underestimated $\tau$ and overestimated $r_e$ values due to the large SEVIRI pixel size. The MODIS 3.7-µm $r_e$ values are consistently lower outside the Sc

cloud regime than the 1.6- and 2.1-µm band retrieved $r_e$, likely because the latter channels are more strongly

influenced by cloud heterogeneity and associated 3D radiative effects in broken clouds.

Figure 7 depicts the regional distribution of two-year mean SEVIRI and MODIS liquid fractional cloud

cover and their differences for all-sky case. The LCF varied between 60 – 100 % within the identified Sc regime in

both datasets; the difference in cloud fraction is within ± 5 %, with SEVIRI being smaller near the coast and larger

further offshore.  SEVIRI retrieves 5-10 % larger LCF in the mid-Atlantic, which also coincides with the largest $H_\sigma$,

indicating the occurrence of the most heterogeneous clouds. The typical liquid clouds here are small broken Cu

leading to larger LCF estimates at the larger SEVIRI pixel size. Around the equator, in contrast, SEVIRI

considerably underestimates MODIS LCF by about 10-30 %. We speculate that the frequent occurrence of ice phase

clouds (deep convection, Ci) here results in overestimating the ice cloud fraction and thus underestimating the liquid

cloud fraction, at the large SEVIRI pixel size. As expected, in the overcast only scenes (LCF ≥ 95 % and $\tau > 3$)

SEVIRI and MODIS LCF agree within ±1 % (not shown).

The daytime-averaged two-year and seasonal statistics of SEVIRI and MODIS LWP are listed in Table 2,

whereas the respective mean $\tau$ and $r_e$ are listed in Table S1. For overcast marine Sc clouds the two-year mean LWP

is 80 g m$^{-2}$ for SEVIRI, 84 g m$^{-2}$ for MODIS 1.6-µm, 88 g m$^{-2}$ for MODIS 2.1-µm, and 87 g m$^{-2}$ for MODIS 3.7-µm

channels. The differences in retrieved LWP values vary from 4 to 8 g m$^{-2}$ (5–10 %), whereas the differences in root

mean square deviation (RMSD) values vary between 16–20 g m$^{-2}$. The SEVIRI and MODIS LWP retrievals are

highly correlated, with correlations > 0.9. In the aerosol-unaffected seasons of DJF and MAM, the difference

between SEVIRI and MODIS LWPs is within 0–5 %. In the heavily polluted months of JJA, LWP retrievals from

SEVIRI are about 10 % lower than those from the MODIS 1.6-µm and 20 % lower than those from the MODIS 3.7-

µm band. This again suggests that SEVIRI retrievals are more strongly affected by the presence of absorbing

aerosols in the Sc regime than the corresponding MODIS 1.6-µm retrievals and that these polluted scenes are not

sufficiently filtered out by the OMI AI threshold. Indeed, unlike in other seasons, in JJA MODIS LWP$_{1.6-µm}$ <

MODIS LWP$_{2.1-µm}$ < MODIS LWP$_{3.7-µm}$, hinting at the influence of absorbing aerosols on MODIS LWP retrievals as

the 3.7-µm channel is known to be the least affected by smoke. In SON, since September is the only month strongly

affected by aerosols, the comparison of SEVIRI and MODIS LWPs is better, with SEVIRI low biases of 6–12 %.

Figure 8 shows the density scatterplots of SEVIRI versus MODIS 1.6-µm LWP, $\tau$, and $r_e$ in the overcast Sc

region for the study period. Most data points are close to the one-to-one line, but with a SEVIRI low bias; the same

feature is also found in monthly and seasonal results. A low $\tau$ bias of 1 compared to all three MODIS channels, and

a low $r_e$ bias of 1 µm compared to MODIS 1.6-µm and a low $r_e$ bias of 1.5 µm compared to MODIS 2.1-µm and 3.7-µm bands are observed in SEVIRI overcast retrievals. The frequency histograms of SEVIRI minus MODIS CPP difference, as well as the differences with respect to different MODIS channels are shown for the all-sky case in Fig. S3 and for the overcast case in Fig. S4. The peak of the LWP absolute difference as well as the relative difference distribution is centred on zero, although the distribution is negatively skewed. Interestingly, in the all-sky case ~40 % of the data have shown negligible difference (zero LWP bias bin), whereas only about 30 % of the data have shown a negligible difference in the overcast case. Histograms of both $\tau$ and $r_e$ differences reveal that the distribution is off centered. Histograms of $\tau$ differences reveal a narrow distribution, which peaks at -1 in the overcast case; however, in the all-sky case there is a broader peak between -1 and 0. Histograms of $r_e$ differences reveal wider distributions, especially when compared against the 2.1- and 3.7-µm channels, which peak at -1 µm in the overcast case; however, in the all-sky case the peak is again broader between -2 and -1 µm.

**4.3 Diurnal cycle of SEVIRI, TMI, and MODIS cloud properties**

Figures 9-11 and S5-S8 show the two-year mean and seasonal diurnal cycle of Sc cloud properties. The diurnal cycles shown here are limited to cases with AI values lower than 0.1, in order to minimize VIS/NIR retrieval biases due to biomass burning smoke (see section 4.1). Because SEVIRI retrievals (black standard and green adiabatic) are only available during daytime, TMI LWP is shown separately for day (red) and night (gray) observations, which combined depict the entire 24-hour diurnal cycle. As before, the analysis is done separately for the all-sky case (solid lines with open circles) and the overcast case (dash-dotted lines with plus signs). MODIS Terra and Aqua values at 10h LST and 14h LST, respectively, are plotted as light blue color symbols (open circles or plus signs) in both cases, for the common 1.6-µm water-absorbing channel, but we report on MODIS $r_e$ for all three absorption channels in the diurnal cycle as they hint at distinct features of cloud heterogeneity and retrieval artifacts.

For the two-year means (Fig. 9), both TMI and SEVIRI indicate a maximum LWP at 06h LST in the morning before sunrise, followed by a decrease until about 16h LST and an increase afterwards. During the night LWP continues to increase until sunrise, as indicated by the TMI night retrievals. At around 06h LST the two-year-mean all-sky LWP values are ~75 g m$^{-2}$ for both TMI and SEVIRI, but they decrease to ~40 g m$^{-2}$ by ~14h LST. This decrease in LWP is linked to a sharp decline in $\tau$ from 11.5 to 5.5 as well as a 20 % decrease in mean fractional cloud cover. The relative variation in $r_e$ is much smaller than that in $\tau$ during the day, in agreement with Zuidema and Hartmann (1995). The $r_e$ values increased by 2 µm in the early hours between 06h and 10h LST, stayed around

11.0-11.5 μm most of the day, and decreased by ~1 μm in the late afternoon by 18h LST. As a result, the diurnal cycle of LWP was mainly driven by τ. Note that the all-sky two-year-mean TMI (red solid line circles), SEVIRI (black solid line circles), and MODIS (light blue circles) LWPs exhibit excellent agreement not only in their relative diurnal variations but also in their absolute values –the curves almost completely overlap.

For the overcast case, a ~30 % increase in τ and a slight <1 μm decrease in $r_e$ lead to an overall increase of 25–30 g m$^{-2}$ (~40 %) in mean LWP compared to the all-sky case. Apart from that, the diurnal cycles of LWP, τ, and $r_e$ are very similar between the overcast and all-sky cases. The standard SEVIRI (black dash-dot line, plus signs) and TMI day (red dash-dot line, plus signs) overcast LWPs also show very good quantitative agreement, with SEVIRI being biased high only about 5 g m$^{-2}$. Note that for the two-year means, adiabatic SEVIRI LWPs (green dashed line, triangles) had larger and negative biases than standard SEVIRI retrievals. As shown later, this was the consequence of the significant smoke-induced negative biases in SEVIRI retrievals in the aerosol-affected seasons of JJA and SON. In the smoke-free seasons of DJF and MAM, adiabatic SEVIRI LWPs were in better agreement with TMI microwave LWPs than were standard SEVIRI LWPs, echoing the findings of Bennartz (2007) and Seethala and Horváth (2010) for MODIS – AMSR-E LWP comparison.

Comparing MODIS Terra (10h LST) and Aqua (14h LST) LWPs, a similar decreasing diurnal trend can be observed, except that MODIS LWPs are 5–10 g m$^{-2}$ larger than SEVIRI LWPs for the overcast case, due to a more pronounced smoke effect (i.e. larger negative τ biases) in the SEVIRI 0.63-μm channel than in the MODIS 0.86-μm channel and also due to the larger SEVIRI pixel size (3 km vs. 1 km). The CM SAF (2016) validation report also suggests that the coarser resolution of SEVIRI retrievals results in somewhat lower τ and LWP values compared to MODIS, because of non-linear averaging effects (plane-parallel albedo bias).

Our results are consistent with Wood et al. (2002) and Painemal et al. (2012), who studied the diurnal variation of LWP over the southeast Atlantic and southeast Pacific Sc, based on microwave and near-infrared satellite data. Similar to our results, Painemal et al. (2012) also noted that τ rather than $r_e$ explains most of the LWP variation. Blaskovic et al. (1990) associated the daytime decrease of LWP with the decrease of cloud geometric thickness observed in their ground-based measurements, as the cloud base height increased from sunrise till mid-afternoon, while cloud top height decreased in the late afternoon. Duynkerke et al. (2004) found that the diurnal variation of Sc LWP is related to the transition from a decoupled MBL during daytime to a vertically well-mixed MBL during the night. The observed diurnal cycle of Sc is characterized by a cloud layer that gradually thickens during the night but gets thinner during the day due to absorption of shortwave radiation and decoupling. The latter

state exhibits slightly negative buoyancy fluxes and a minimum vertical velocity variance near cloud base. This implies that surface-driven, moist thermals cannot penetrate the cloud layer, while entrainment maintains a steady supply of relatively warm and dry air from just above the inversion into the cloud layer, resulting in a distinct LWP diurnal cycle with minimum values during the day. The diurnal cycle of LWP also consistently follows the variation of cloud fraction in our data, as shown in Figs. 10 and S8. This is in agreement with Fairall et al. (1990) and Ciesielski et al. (2001), who observed that fractional cloudiness is maximum in the predawn hours and minimum in the mid-afternoon, which is accompanied by an opposite trend in the MBL moisture with a predawn drying and an afternoon moistening.

The seasonal mean diurnal cycles of Sc clouds are qualitatively similar to the two-year mean, except for the aerosol affected months of JJA (Figs. 11 and S5-S8). The maximum LWP tends to occur between 06h and 10h LST. The largest diurnal variation is seen during SON, which is also the season with the greatest cloud cover. We found that the relative amplitude of the two-year- and seasonal-mean LWP diurnal cycle is typically 35-40 %. Wood et al. (2002) reported diurnal amplitudes of 15-35 % in MBL clouds using TMI data and Zuidema and Hartmann (1995) obtained a 25 % variation in LWP over the North/South Pacific as well as South Atlantic stratus clouds using SSM/I data for the summer months. However, Fairall et al. (1990) found larger amplitudes of 60-70 % for Californian Sc clouds using a 17-day period of near-continuous ground-based microwave radiometer data.

In the all-sky case, the diurnal variation of TMI and SEVIRI LWP is in good absolute agreement within $\pm 5$ g m$^{-2}$, for all seasons and the two-year mean, except JJA. In JJA, however, a $\pm 10$ g m$^{-2}$ or even slightly larger mean difference is found between the techniques, despite the exclusion of aerosol-affected pixels with AI > 0.1. MODIS Terra and Aqua mean LWPs also show excellent agreement with the corresponding SEVIRI LWPs within $\pm 5$ g m$^{-2}$, for all seasons and the two-year mean.

In the overcast case, SEVIRI LWPs are 10–20 g m$^{-2}$ larger than TMI LWPs especially for the aerosol-free seasons of DJF and MAM. After applying the adiabatic correction, the biases become negligible between the datasets. For the aerosol-affected seasons of JJA and SON, the mean SEVIRI LWPs likely underestimate the actual values and hence applying adiabatic correction (reduction) worsens the comparison with TMI LWPs. In the overcast case, MODIS Terra and Aqua LWPs deviate by 5–10 g m$^{-2}$ from SEVIRI LWPs for the aerosol-free seasons, but by a larger amount of 5–20 g m$^{-2}$ for the aerosol-affected seasons due to smoke-induced biases being larger in SEVIRI than MODIS retrievals (see section 4.2).

Seasonally, $\tau$ varies between 4 and 16, typically showing a relative decrease of 40–50 % from early morning to late afternoon. Not surprisingly, the diurnal amplitude of $\tau$ is similar to that of LWP. Although the absolute value of $r_e$ varies from 7 to 12 µm between different seasons, the relative diurnal variation is negligible. The diurnal reduction in $\tau$ is likely due to the reduction in cloud fraction and cloud geometric thickness, while the variation in cloud-top $r_e$ is probably indicative of enhanced cloud-top entrainment of dry air and associated droplet

evaporation. Although MODIS $\tau$ values are slightly higher in both Aqua and Terra data, the difference with SEVIRI is only about 1, while MODIS $r_e$ values are within 2 µm of the SEVIRI $r_e$. Interestingly in the all-sky case, MODIS 1.6- and 2.1-µm $r_e$ are 1–2 µm larger than the SEVIRI $r_e$, and the MODIS 3.7-µm $r_e$ agrees best with SEVIRI. This suggests that in lower cloud fraction, more heterogeneous scenes, the MODIS 1.6- and 2.1-µm channels retrieve larger values (plane-parallel $r_e$ bias), whereas in SEVIRI the applied climatological weighting lowers the actual $r_e$

values and makes them more comparable to the MODIS 3.7-µm $r_e$, which are least affected by cloud heterogeneity and 3D effects.

### 5. Summary

The objective of this work was to compare LWP, $\tau$, and $r_e$ retrievals from SEVIRI and MODIS, and LWP from

TMI, in order to quantify the effect of biomass burning smoke on passive VIS/NIR imager retrievals as well as to evaluate the diurnal cycle of South Atlantic marine stratocumulus clouds. In general, SEVIRI and TMI showed good agreement for instantaneous and domain-mean LWPs in the extensive Sc region, while the agreement in broken clouds was worse. Spatial distributions showed a correlation higher than 0.85 between the all-sky retrievals, with negligible bias on a two-year and seasonal basis for all smoke-free months. Austral winter months were heavily

smoke-affected and hence a larger bias was observed between the VIS/NIR and microwave techniques, due to an underestimation in the former. For overcast cases, the mean LWPs were ~60 % greater than the all-sky LWPs in both SEVIRI and TMI. In biomass smoke-free months, the overcast SEVIRI LWPs were higher than the corresponding TMI LWPs; however, an adiabatic correction could reduce this high bias to the 5 % level. In smoke-affected months, in contrast, the adiabatic correction, which amounts to a ~17 % reduction in VIS/NIR LWP, further

increased the (negative) bias between SEVIRI and TMI. This was so because SEVIRI retrievals were already biased low by the presence of absorbing aerosols over clouds, even though aerosol index-based filtering was applied in an attempt to exclude the most polluted pixels.

SEVIRI and MODIS LWPs showed excellent correlation of > 0.9 in the Sc region on a two-year and seasonal basis. However, mean MODIS $\tau$ and $r_e$ were both 5–10 % higher in smoke-free months and 10–20 %

higher in smoke-affected months than corresponding SEVIRI mean values. Interestingly, in overcast cases the relative magnitudes of MODIS $r_e$ retrievals from the 1.6-, 2.1-, and 3.7-µm channels were different in smoke-free and smoke-affected months. Especially in JJA, the 1.6-µm MODIS $r_e$ was significantly lower than the other two values, indicating a strong low bias in this channel due to smoke absorption. Overall, the difference between SEVIRI and MODIS LWPs was within 5 % in smoke-free seasons, but the retrieved SEVIRI LWPs were 10–25 %

lower than MODIS LWPs in smoke-affected months.

Prompted by the above, we separately investigated the influence of absorbing aerosols over the Sc domain using aerosol index obtained from OMI. While TMI LWP showed a steep increase with AI, SEVIRI LWP showed a systematic decrease. This indicates that absorbing aerosols above liquid clouds introduce substantial negative retrieval biases in VIS/NIR cloud optical thickness and droplet effective radius, and, hence, in the deduced LWP.

This bias in SEVIRI LWP increased with AI and could be as large as 40 g m$^{-2}$ in instantaneous retrievals. Neglecting aerosol-affected pixels with AI > 0.1, the domain-mean TMI minus SEVIRI LWP bias could be reduced but not completely removed. Similar to SEVIRI, all three MODIS channels showed a decrease in LWP with AI, with the largest decrease occurring in the 1.6-µm channel. The overall reduction in LWP with AI was 10–20 % in MODIS retrievals, whereas it was ~50 % in SEVIRI retrievals. The larger sensitivity of SEVIRI retrievals to smoke can

partially be explained by the difference in the non-absorbing VIS channel employed in the bispectral methods: the 0.86-µm channel used for MODIS oceanic retrievals is less affected by aerosol absorption than the 0.63-µm channel used for SEVIRI.

Our finding that SEVIRI $\tau$ and $r_e$ were both lower than their (1.6-µm) MODIS counterparts, however, is puzzling. In the absence of considerable net 3D effects, subpixel heterogeneity within the larger SEVIRI footprint

would alone lead to a simultaneous underestimate of $\tau$ (plane-parallel albedo bias) and overestimate of $r_e$ (plane-parallel effective radius bias; Zhang et al., 2012) compared to the higher resolution MODIS retrievals. Indeed, such opposite-sign biases were found by Painemal et al. (2012) in GOES-10 $\tau$ and $r_e$ compared to MODIS data in the southeast Pacific marine Sc region. Therefore, the SEVIRI $\tau$ underestimation is consistent with plane-parallel biases, while the SEVIRI $r_e$ underestimation is not.

We performed a preliminary analysis of factors that might explain these biases. We investigated the variation of gridbox-mean SEVIRI and MODIS $\tau$, $r_e$, and LWP with heterogeneity, using $H_\sigma$ (the normalized

standard deviation of SEVIRI 0.63-µm reflectances) to characterize the inhomogeneity of an overcast $0.25^o$ x $0.25^o$ gridbox. Similar to Painemal et al. (2013b), we found both for SEVIRI and for MODIS a decrease in $\tau$ and increase in $r_e$ with increasing scene heterogeneity under constant TMI LWP. However, the negative SEVIRI-MODIS $r_e$

difference remained a robust feature, independent of the magnitude of scene heterogeneity. A more promising potential contributor to the negative SEVIRI-MODIS $r_e$ bias is the difference between the SEVIRI and MODIS view geometries. The view geometry (view zenith and azimuth angle) of the geostationary SEVIRI is fixed for a given geographic location. The polar-orbiter MODIS view geometry of the same geographic location, however, depends on the cross-track position of that location within the MODIS swath and varies from orbit to orbit. A preliminary

analysis indicated view zenith angle (VZA) dependence in the SEVIRI-MODIS $r_e$ bias, with the bias varying between -0.5 µm to -2 µm and generally being lowest for the MODIS oblique backscatter view direction. In addition to the effect of different view geometries, we speculate that the $r_e$ low bias is also related to algorithmic and spectral differences between MODIS and SEVIRI (e.g. non-identical look-up-tables, different sensitivities to absorbing aerosol, ozone, and Rayleigh scattering in the visible channels used over ocean, residual calibration issues). All

these potential contributing factors to the negative SEVIRI $r_e$ bias will be thoroughly investigated in a future study.

In the all-sky case, the diurnal variations of TMI and SEVIRI LWP were in good absolute agreement, being within ±5 g m$^{-2}$ for all seasons and the two-year mean, except JJA. In JJA, the season most affected by biomass smoke, a larger mean difference was found between the techniques, although we made an attempt to eliminate aerosol-affected pixels with AI > 0.1. MODIS Terra and Aqua mean LWPs also showed excellent agreement with

corresponding SEVIRI LWPs in the all-sky case, differences being within ±5 g m$^{-2}$ for all seasons and two-year means.

In the overcast case, SEVIRI LWPs were 10–20 g m$^{-2}$ larger than TMI LWPs especially in the smoke-free seasons of DJF and MAM. After applying an adiabatic correction to SEVIRI retrievals, however, the biases between the datasets became negligible. In the smoke-affected seasons of JJA and SON, the mean SEVIRI LWPs already

underestimated the TMI values due to smoke-induced retrieval biases and hence applying the adiabatic correction (i.e. further reduction) worsened the comparison with TMI LWPs. In the overcast case, MODIS Terra and Aqua LWPs differed by 5–10 g m$^{-2}$ from SEVIRI LWPs in smoke-free seasons and by a larger amount of 5–20 g m$^{-2}$ in smoke-affected seasons, due to the different magnitudes of smoke-induced biases in SEVIRI and MODIS retrievals.

Irrespective of season, both TMI and SEVIRI LWP decreased from morning to mid-afternoon, and after

that a slow increase was observed. Clouds are the thickest prior to sunrise and as the day progresses the cloud layer

thins due to the absorption of solar radiation and associated decoupling of the sub-cloud layer. We found that the relative amplitude of the LWP diurnal cycle is typically 30-50 %, which is close to but slightly larger than the diurnal amplitude reported in most previous studies. The diurnal variation in SEVIRI LWP was mainly due to that in cloud optical thickness, while droplet effective radius showed relatively small diurnal variability. MODIS Terra

(morning) and Aqua (afternoon) LWPs indicated a similar diurnal trend, but MODIS LWPs were 5–10 g m$^{-2}$ larger than SEVIRI/TMI values in the overcast case. This was maybe partly due to the plane-parallel albedo bias affecting the larger SEVIRI pixels.

While the discrepancies between microwave and VIS/NIR LWP retrievals in areas of broken clouds with low cloud fraction require further research to be fully resolved, our study has shown that there is a reasonable

consensus between the techniques about the seasonal and diurnal cycles of LWP in nearly overcast stratocumulus fields. This lends some credibility to the VIS/NIR retrievals of the underlying cloud microphysical properties. In our opinion, SEVIRI-derived CLAAS-2 cloud property observations provide a useful resource for the evaluation of stratocumulus diurnal cycles in climate models.


**Acknowledgements**

The work of the first author was funded by EUMETSAT through the CM SAF Visiting Scientist scheme and was completed at the Royal Netherlands Meteorological Institute, De Bilt, Netherlands. SEVIRI CLAAS-2 CPP retrievals were obtained from https://doi.org/10.5676/EUM_SAF_CM/CLAAS/V002. TMI data were produced by Remote Sensing Systems and sponsored by the NASA Earth Sciences Program. Data are available at www.remss.com/missions/tmi. OMI data were downloaded from http://mirador.gsfc.nasa.gov/ and Aqua/Terra MODIS Level-2 data were obtained from http://ladsweb.nascom.nasa.gov/. The first author acknowledges Dan Cayan at Scripps Institution of Oceanography, La Jolla, California, for providing access to his computational facility to finish a part of this work. We also thank two anonymous reviewers for their insightful comments and suggestions, which greatly improved our manuscript.

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

**TABLES**

**Table 1.** Two-year mean and seasonal statistics of collocated SEVIRI and TMI LWP retrievals for rain-free, ice-free, smoke-free (AI < 0.1), $\tau$ > 3, and overcast (LCF ≥ 95%) grid cells over the marine stratocumulus region. LWP means, biases (SEVIRI-TMI), and Root Mean Square Differences (RMSD) are given in g m$^{-2}$. The $\tau$ means and $r_e$ means (in micron) are also tabulated. The values in brackets are statistics without filtering for LCF ≥ 95% and $\tau$ > 3, i.e., for the all-sky case. "adb" refers to the overcast LWP calculation assuming adiabatic clouds.

|  | JJA | SON | DJF | MAM | Two-year |
|---|---|---|---|---|---|
| *Stratocumulus (SEVIRI vs. TMI)* | | | | | |
| SEVIRI LWP | 73 (48) | 87 (63) | 92 (52) | 83 (41) | 84 (53) |
| SEVIRI LWP adb | 61 | 72 | 76 | 69 | 70 |
| TMI LWP | 82 (57) | 82 (62) | 76 (45) | 73 (39) | 80 (53) |
| SEVIRI-TMI LWP | -9 (-9) | 5 (1) | 16 (7) | 10 (2) | 4 (0) |
| SEV adb -TMI LWP | -21 | -10 | 0 | -4 | -10 |
| RMSD | 31 (26) | 28 (26) | 22 (21) | 21 (20) | 28 (24) |
| #Samples | 1.6E+5 (3.2E+5) | 3.3E+5 (5.2E+5) | 1.4E+5 (3.2E+5) | 9.1E+4 (2.6E+5) | 7.3E+5 (1.4E+6) |
| Correlation | 0.81 (0.87) | 0.86 (0.89) | 0.92 (0.93) | 0.93 (0.92) | 0.86 (0.89) |
| SEVIRI $\tau$ | 10.2 (6.8) | 11.0 (8.2) | 10.7 (6.6) | 10.3 (5.7) | 10.7 (7.0) |
| SEVIRI $r_e$ | 9.4 (10.8) | 10.6 (11.2) | 11.6 (11.7) | 10.8 (11.0) | 10.6 (11.2) |

**Table 2.** Two-year mean and seasonal statistics of collocated SEVIRI and MODIS retrievals in rain-free, ice-free, smoke-free (AI < 0.1), $\tau > 3$, and overcast (LCF ≥ 95%) grid cells over the marine stratocumulus region. LWP means, biases (MODIS-SEVIRI), and Root Mean Square Differences (RMSD) are given in g m$^{-2}$. Corresponding $\tau$ means and $r_e$ means are tabulated in Table S1. The values in brackets are statistics without filtering for LCF ≥ 95% and $\tau > 3$, i.e., for the all-sky case. "S" and "M" refer SEVIRI and MODIS.

| | JJA | SON | DJF | MAM | Two-year |
|---|---|---|---|---|---|
| Stratocumulus (SEVIRI vs. MODIS) | | | | | |
| SEVIRI LWP | 71 (44) | 81 (58) | 88 (45) | 81 (39) | 80 (48) |
| MODIS 1.6 LWP | 79 (46) | 86 (59) | 88 (45) | 84 (41) | 84 (49) |
| MODIS 2.1 LWP | 85 (48) | 90 (60) | 89 (44) | 85 (40) | 88 (50) |
| MODIS 3.7 LWP | 88 (51) | 89 (59) | 84 (42) | 82 (38) | 87 (49) |
| M 1.6 – S LWP | 8 (2) | 5 (1) | 0 (0) | 3 (2) | 4 (1) |
| M 2.1 – S LWP | 14 (4) | 9 (2) | 1 (-1) | 4 (1) | 8 (2) |
| M 3.7 – S LWP | 17 (7) | 8 (1) | -4 (-3) | 1 (-1) | 7 (1) |
| RMSD 1.6 | 17 (21) | 16 (20) | 17 (18) | 16 (18) | 17 (19) |
| RMSD 2.1 | 16 (21) | 15 (19) | 17 (18) | 16 (19) | 16 (19) |
| RMSD 3.7 | 19 (20) | 19 (20) | 18 (19) | 18 (17) | 20 (19) |
| #Samples | 3.7E+5 (9.6E+5) | 6.4E+5 (1.4E+6) | 2.4E+5 (8.2E+5) | 2.0E+5 (7.8E+5) | 1.5E+6 (4.0E+6) |
| Corr. LWP 1.6 | 0.95 (0.91) | 0.95 (0.93) | 0.95 (0.94) | 0.96 (0.94) | 0.95 (0.93) |
| Corr. LWP 2.1 | 0.95 (0.91) | 0.95 (0.93) | 0.95 (0.94) | 0.96 (0.93) | 0.95 (0.93) |
| Corr. LWP 3.7 | 0.94 (0.92) | 0.93 (0.92) | 0.94 (0.93) | 0.95 (0.95) | 0.93 (0.93) |



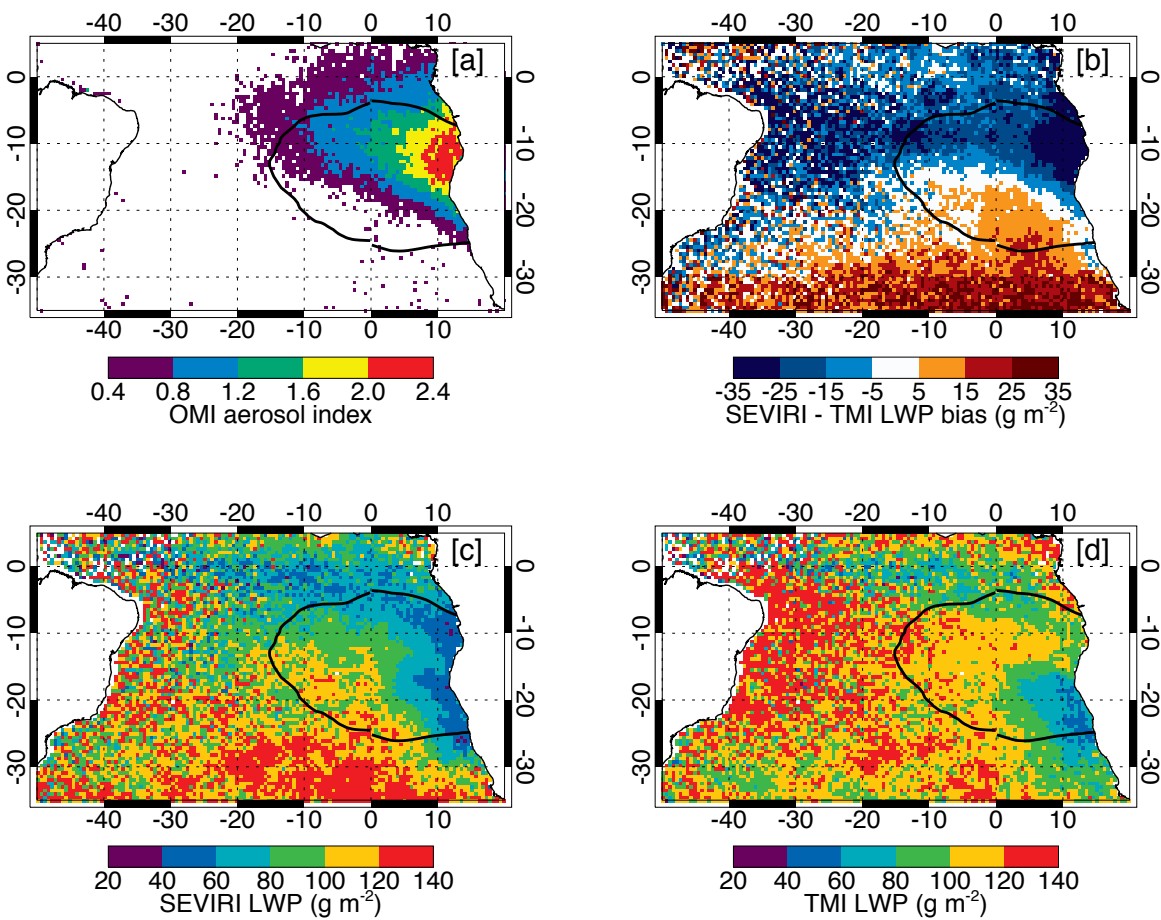



**Figure 1**. Spatial distribution of (a) OMI ultraviolet aerosol index, (b) SEVIRI minus TMI liquid water path bias, (c) SEVIRI

liquid water path, and (d) TMI liquid water path, averaged for JAS in 2011 and 2012 for overcast (LCF ≥ 95 % and $\tau$ > 3) rain-

and ice-free conditions. The black contour denotes the identified stratocumulus region.

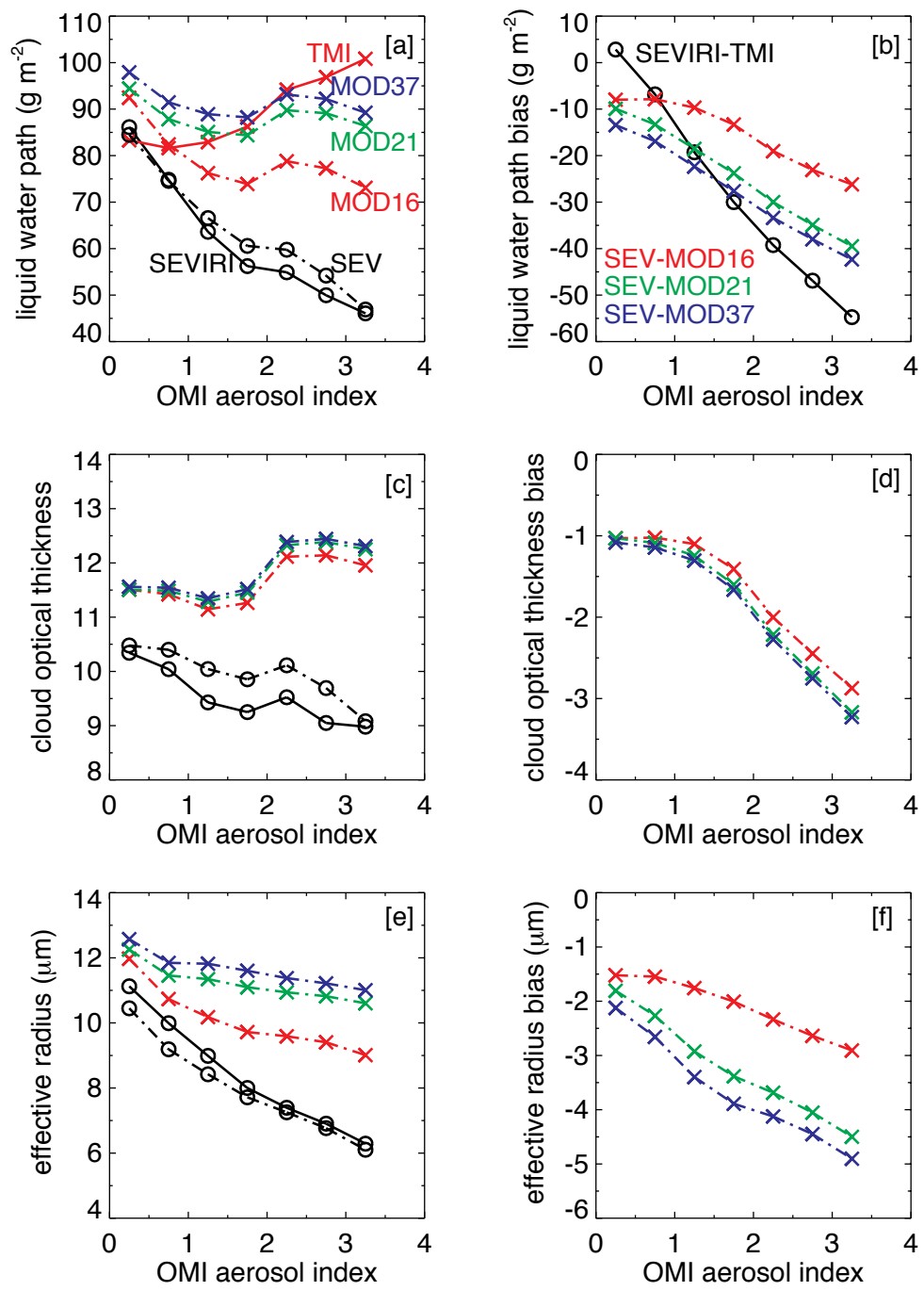


**Figure 2**. OMI aerosol index *versus* (a) SEVIRI, TMI and MODIS LWPs, (b) SEVIRI LWP biases compared to TMI and MODIS, (c) SEVIRI and MODIS $\tau$, (d) SEVIRI – MODIS $\tau$ biases, (e) SEVIRI and MODIS $r_e$, and (f) SEVIRI – MODIS $r_e$

biases, over the overcast Sc region for JAS 2011 and JAS 2012 for rain- and ice-free conditions. Solid lines correspond to SEVIRI vs. TMI comparison, whereas dash-dotted lines correspond to SEVIRI vs. MODIS comparison. The label "SEV" refers SEVIRI values at MODIS collocations.


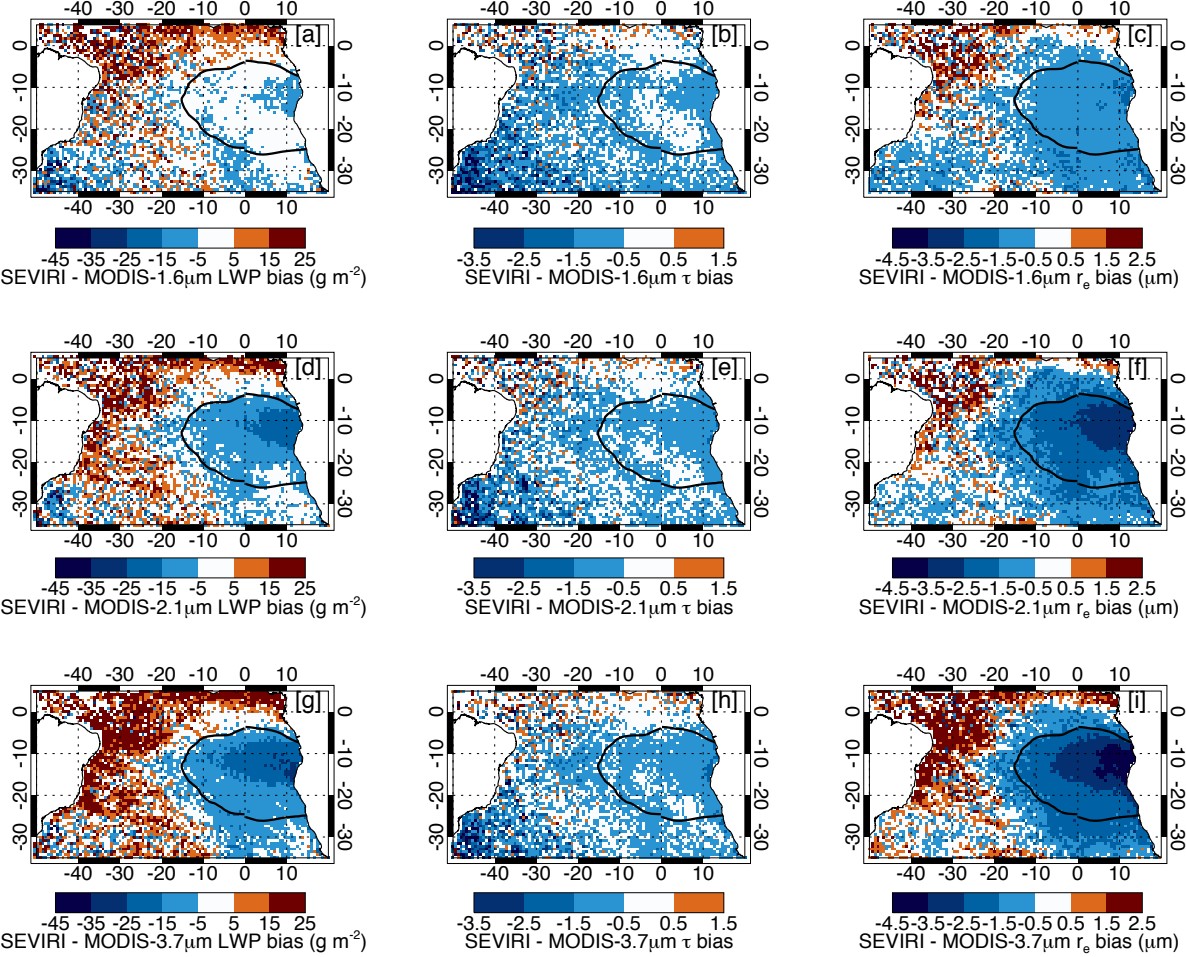

**Figure 3**. Spatial distribution of SEVIRI liquid water path biases (a, d, g), cloud optical thickness biases (b, e, h), and droplet effective radius biases (c, f, i), compared to MODIS 1.6-, 2.1-, and 3.7-µm channel retrievals, respectively, averaged for JAS 2011 and JAS 2012 for overcast (LCF ≥ 95 % and τ > 3) rain- and ice-free conditions.




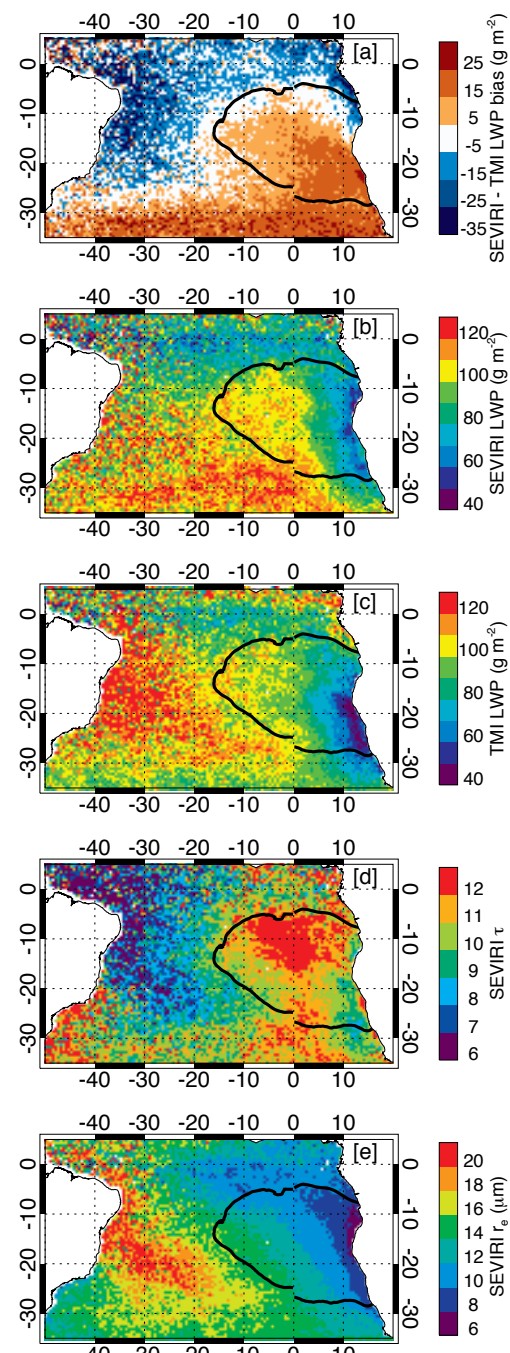


**Figure 4.** Two-year mean map of (a) SEVIRI minus TMI LWP bias, (b) SEVIRI LWP, (c) TMI LWP, (d) SEVIRI τ, (e) SEVIRI

1.6-μm $r_e$, for the overcast case (LCF ≥ 95 % and τ > 3). The solid black contour denotes the identified Sc region. Rain-, ice-, and

smoke-free conditions were applied.


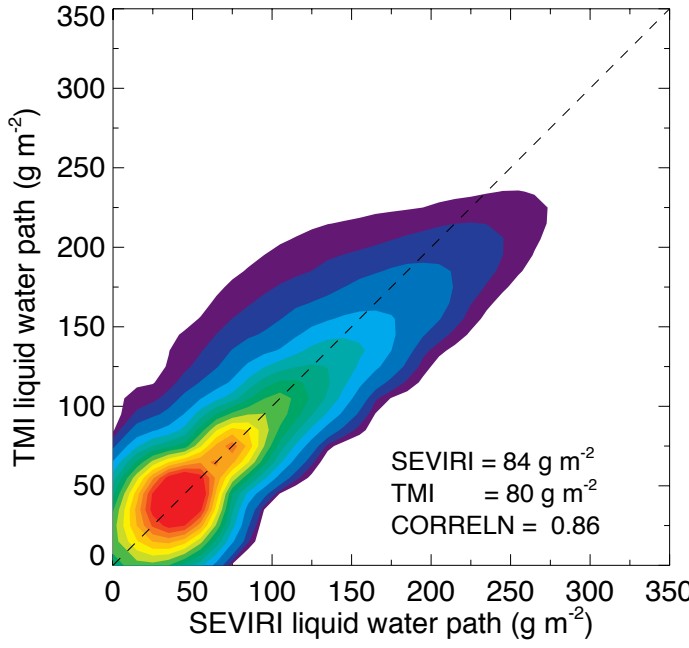

**Figure 5.** Scatter density plot of SEVIRI versus TMI liquid water path for the overcast case (LCF $\geq$ 95 % and $\tau$ > 3) in two years of data. Rain-, ice-, and smoke-free conditions were applied.




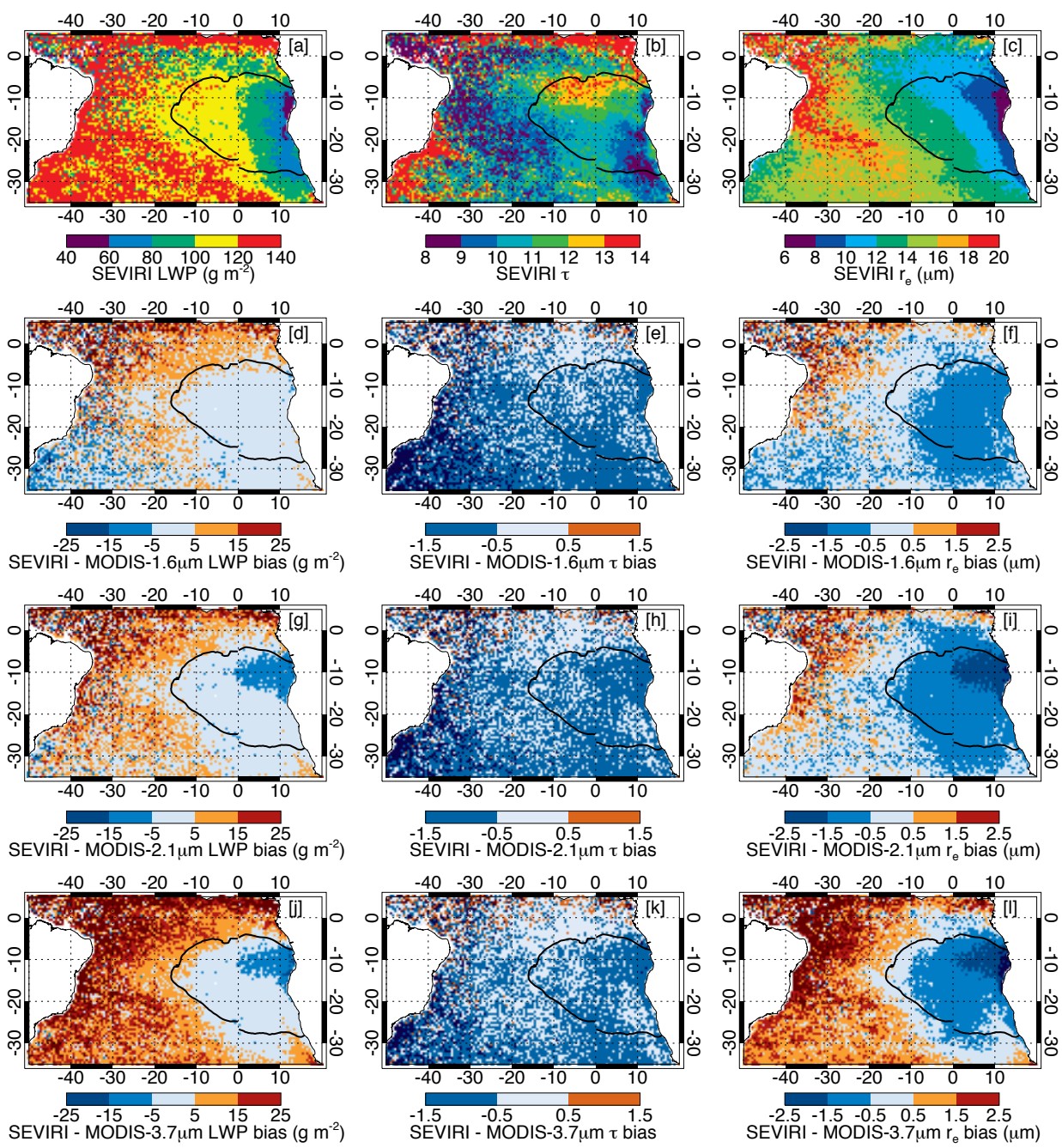

**Figure 6.** Two-year mean map of (a) SEVIRI LWP, (b) SEVIRI $\tau$, (c) SEVIRI $r_e$, (d, g, j) SEVIRI minus MODIS liquid water path biases, (e, h, k) SEVIRI minus MODIS cloud optical thickness biases, (f, i, l) SEVIRI minus MODIS droplet effective radius biases for the overcast case (LCF $\geq$ 95 % and $\tau$ > 3) in ice- and smoke-free conditions.

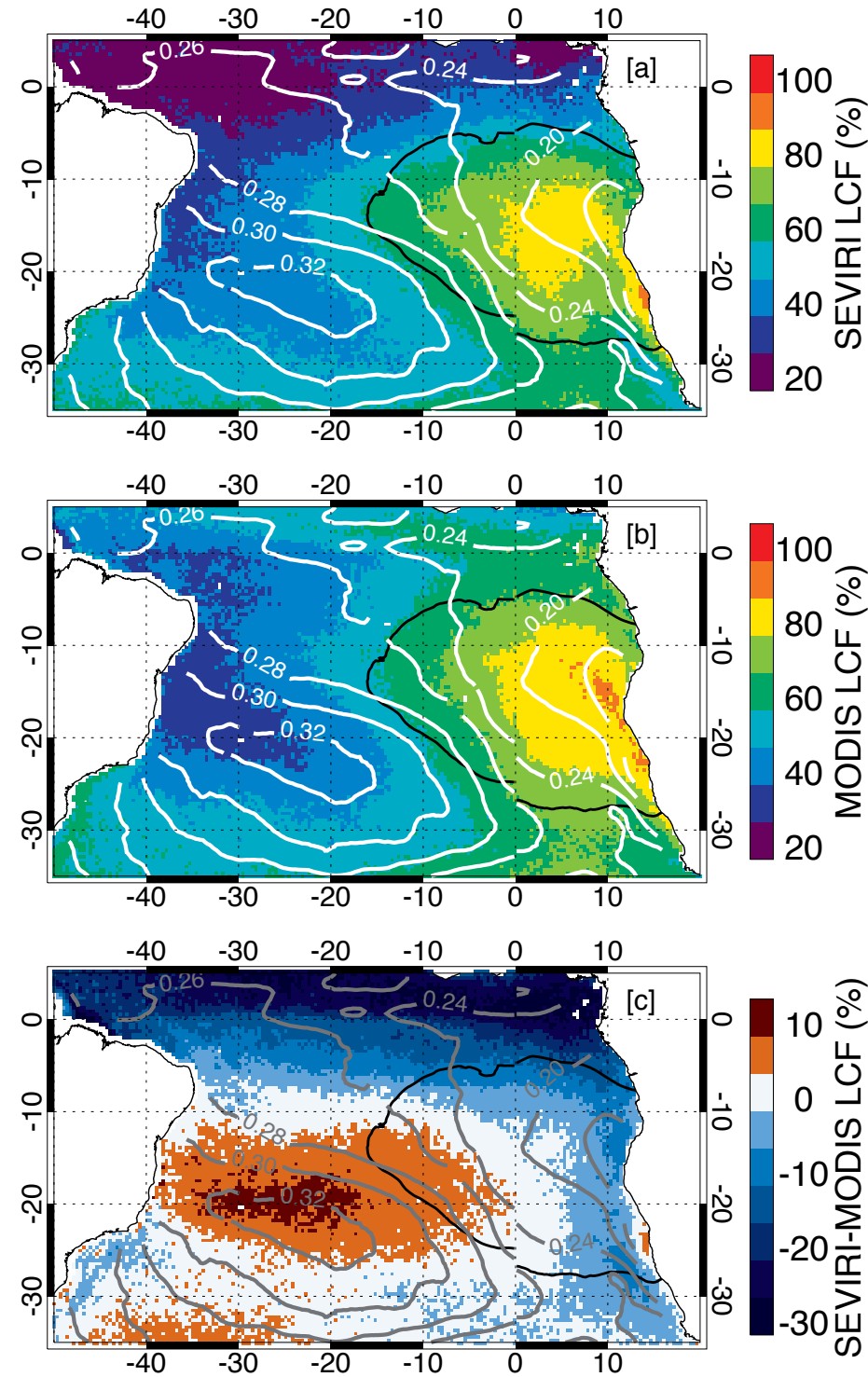

**Figure 7.** Two-year mean map of liquid fraction of cloud amount of (a) SEVIRI, (b) MODIS, and (c) SEVIRI-MODIS bias for

all-sky case. Contours represent the heterogeneity measure H. computed from 3-km resolution SEVIRI 0.63-µm reflectances.


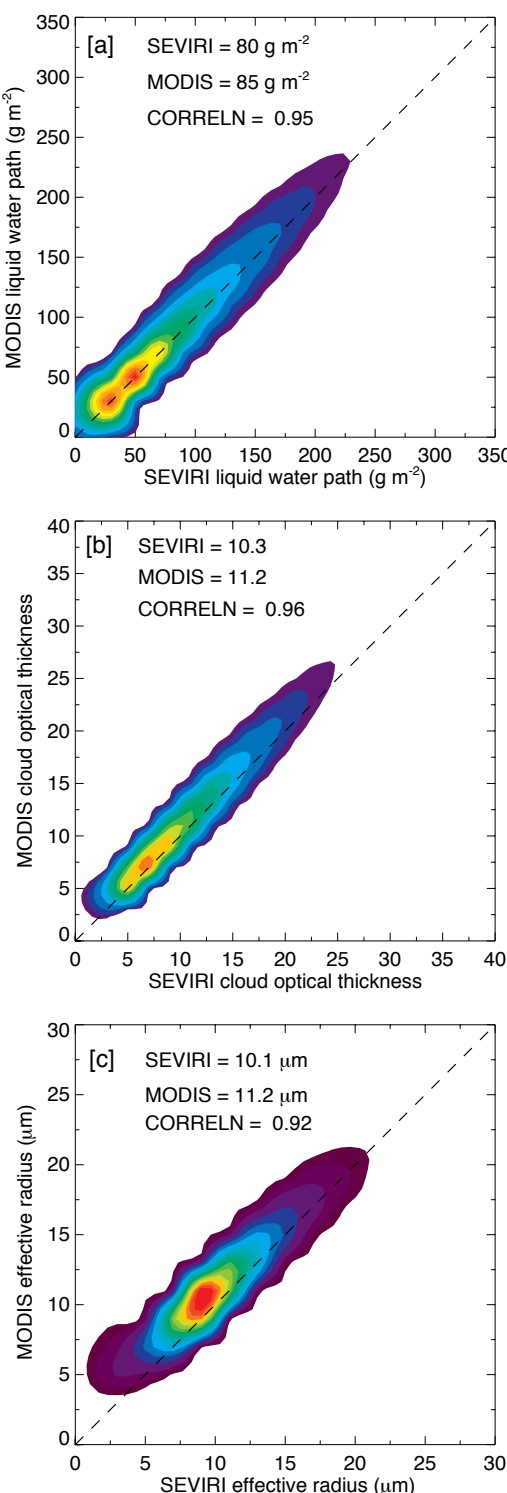


**Figure 8.** Scatter density plot of SEVIRI versus MODIS 1.6-μm liquid water path, cloud optical thickness, and effective radius in the overcast case (LCF ≥ 95 % and τ > 3) in two years of data. Rain-, ice-, and smoke-free conditions were applied.

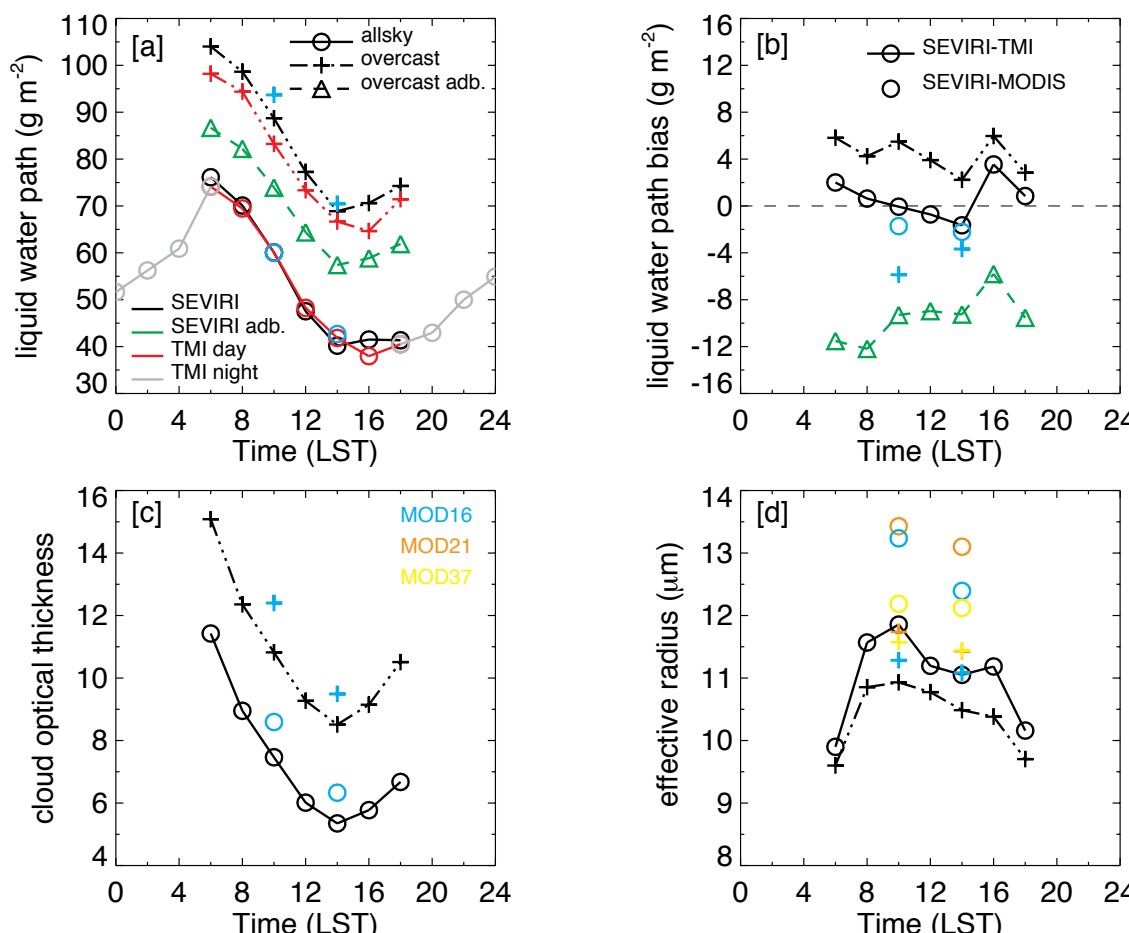



**Figure 9.** Two-year mean diurnal cycle of cloud properties over the Sc region, both for all-sky and overcast (LCF ≥ 95 % and τ >

3) cases: (a) SEVIRI and TMI LWPs, (b) SEVIRI minus TMI LWP bias, (c) SEVIRI τ, and (d) SEVIRI 1.6-μm $r_e$. MODIS Terra

and Aqua values are also plotted. Rain-, ice-, and smoke-free conditions were applied.


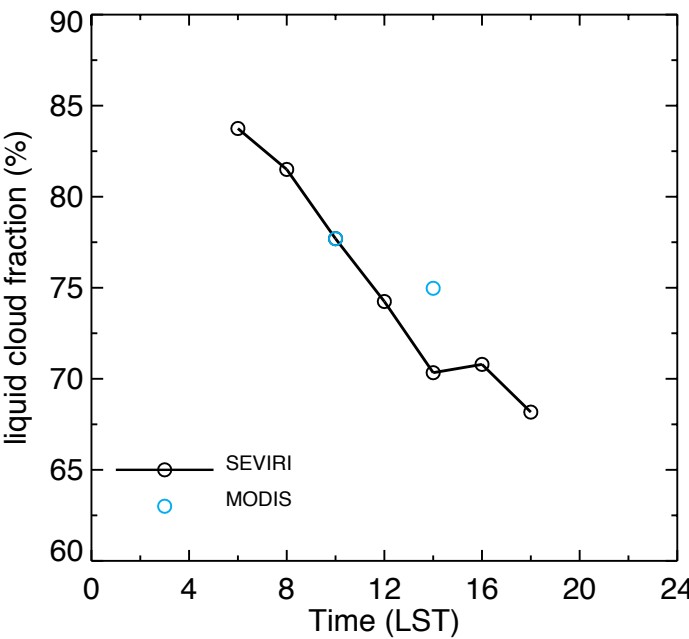




**Figure 10.** Two-year mean diurnal cycle of SEVIRI LCF over the Sc region for the all-sky case. MODIS Terra and Aqua values

are also plotted. Rain-, ice-, and smoke-free conditions were applied.

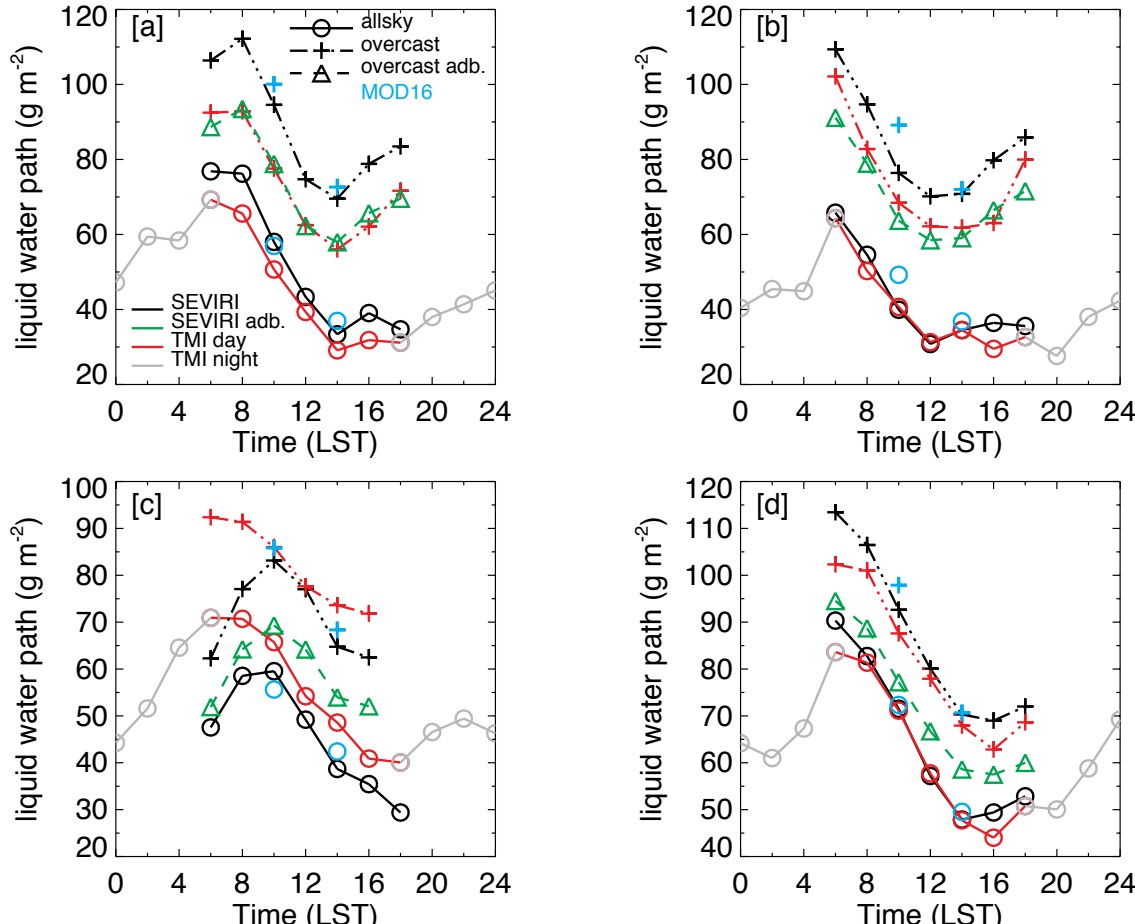

**Figure 11.** Seasonal mean diurnal cycle of SEVIRI and TMI LWPs over the Sc region, both for all-sky and overcast (LCF ≥ 95 % and τ > 3) cases: (a) DJF, (b) MAM, (c) JJA, and (d) SON for December 2010 to November 2012. MODIS Terra and Aqua values are also plotted. Rain-, ice-, and smoke-free conditions were applied.


