# Peer review of "Evaluating the diurnal cycle of South Atlantic stratocumulus clouds as observed by MSG-SEVIRI"

_Atmospheric Chemistry and Physics, 2018_

## Referee Comment (RC1) · Anonymous Referee #1 · 19 Jun 2018

REVIEW Seethala et al. (2018 ACP):

The manuscript documents marine low-cloud properties over the southeast Atlantic Ocean by combining TMI liquid water path and cloud retrievals from MODIS and SEVIRI. The specific focus of the paper is on daytime effective radius (reff), optical thickness (COT), and liquid water path from SEVIRI. The article is relevant as SEVIRI is the only satellite sensors with the necessary temporal resolution to resolve the diurnal cycle in cloud properties. Moreover, the manuscript is one of the few studies that attempt to analyze the diurnal cycle in reff and tau over this cloud regime. While I believe the paper should be accepted for publication, I would like to encourage the authors to refine part of their analysis, provide support to several claims made throughout the manuscript, and clarify the main objective of their study. Since I have coauthored a couple of papers on the same topic, I am offering suggestions based on my past experience with geostationary retrievals (do not feel obligated to cite my papers).

Main objective: It is unclear if the goal of the study is to understand the seasonal/diurnal cloud evolution of the Namibia-Angola stratiform cloud regime (as suggested by the title) or to evaluate SEVIRI cloud retrievals with other datasets (most of the analysis revolves around the differences between SEVIRI and other datasets, and potential bias in SEVIRI retrievals). If the focus is to characterize the diurnal cycle, then please provide more detailed information about the amplitude of the daytime cycle and explain spatial/temporal changes in the context of the atmospheric circulation and thermodynamical structure. For instance, in Painemal et al. (2012, JGR), we attempted to understand the dynamical factors behind the cloud diurnal cycle in the SE Pacific, and showed hourly composites (maps) of cloud retrievals. In Painemal et al. (2013, J. Atmos Sc.), we further endeavored to understand variations in liquid water path and cloud fraction, in the context of the boundary layer depth evolution and subsidence variability. Similarly, we utilized a super-parameterized climate model and NASA-Langley SEVIRI retrievals for describing the diurnal evolution of cloud fraction and height over the Namibia-Angola stratocumulus cloud deck (Painemal et al., 2015 J. Climate). If the focus is mostly evaluating the ability of SEVIRI to reproduce the diurnal cycle, please modify the title and the introduction accordingly. More suggestions for evaluating SEVIRI are provided below.

Inhomogeneity and cloud mask: I was surprised that cloud fraction (mask) differences between SEVIRI CLAAS and MODIS collection 6 were not analyzed. I would speculate that the spatial pattern of the SEVIRI-MODIS difference near the equator in Figure S5d is most likely due to cloud fraction differences between both sensors/algorithms. Also, the use of some sort of inhomogeneity index would provide support to the hypothesis that pixel resolution is in part responsible for the COT difference between SEVIRI and

MODIS. Although the plane parallel bias is likely playing a role, it is puzzling that both SEVIRI COT and reff are generally smaller than their MODIS counterparts (1.6um), as one would expect that reff (COT) is overestimated (underestimated) as the pixel resolution becomes coarser (e.g. table 1a and 1b in Painemal et al., 2012 JGR). This points to other issues associated with differences in the retrieving algorithms, since SEVIRI visible channels were calibrated against MODIS. The question could be answered if the CLAAS algorithm were applied to MODIS (I do not know if this is even possible). At the very least, the authors should speculate about the causes for the inconsistencies between MODIS and SEVIRI that cannot be explained by the plane parallel bias or absorbing aerosols. Lastly, if the COT threshold was applied to SEVIRI (and MODIS), then a comparable threshold should be applied to TMI. If not, the comparison between TMI and SEVIRI.

Inconsistent results: The curve "SEV" in Fig. 2a is SEVIRI collocated with MODIS, correct? Does it mean that not always MODIS and SEVIRI are collocated? All the retrievals should be spatially and temporally collocated. I also noticed that in Fig S9, SEVIRI reff for overcast and all-sky samples is almost the same (Fig. S9a and b). In contrast, the all-sky and overcast averages are quite different for MODIS, why?

What is the purpose of showing the three MODIS reff's as the apples-to-apples comparison is between MODIS-SEVIRI at 1.6 um? I understand that this is useful for understanding the reff bias due to absorbing aerosols (which is not a novel result), but it is extremely confusing to understand figures 6-7 with so many symbols and retrievals, and the overall objective of including MODIS 2.1 and 3.7 reff is unclear. Similarly, just report one MODIS COT as the three MODIS COT are essentially the same. The authors mention that differences in MODIS reff at 1.6, 2.1, and 3.7 um might be providing information about the cloud vertical structure and precipitation; however, numerous papers (e.g. Zhang and Platnick, 2011, Painemal et al. 2013 ACP: "The impact of horizontal heterogeneities, cloud fraction, and liquid water path on warm cloud effective radii from CERES-like Aqua MODIS retrievals) have shown that the difference between

the 3.7 um and 2.1 um reff (or 1.6 um) mostly reflect the effect of spatial inhomogeneity and clear-sky contamination in the retrievals (for stratiform clouds).

Figs S2 and S5 deserve to be included in the manuscript.

Remove "Discussion" from the title of Section 4. Page 12, line 349, I disagree, Figure S1 does not show any difference for COT.

Page 5, first paragraph: This is non-raining pixels according to the RSS algorithm. Depending on the threshold used to define a rainy pixel, drizzle or light precipitation is still possible.

Page 15, line 435: I cannot find the figure that shows the overestimation of SEVIRI (relative to MODIS) for broken scenes.

Page 16, first paragraph. This explanation is unlikely, since you removed samples with high liquid water path (precipitating samples according to the RSS algorithms). Moreover, the spectral difference in reff is mostly indicative of the effect of spatial inhomogeneity and 3D radiative effects in the retrievals rather than information about the vertical structure or precipitation.

Page 17, line 487, what do you mean by "smaller...". Page 19 lines 547-549: I suspect this is mostly due to cloud thinning. If cloud fraction (mask) played a role, then cloud effective radius would be biased high due to clear-sky contamination and 3D radiative effects. Page 20 line 558: Use austral winter instead.

Page 4, line 102, replace "evaluated" with "analyzed". Page 4, line 110, Add "In contrast," before Painemal et al. (2012). Painemal et al. (2012) also utilized in-situ cloud probe to assess the bias in satellite cloud properties. Page 5 line 143: Define VIS/NIR. Page 6, line 157, If the physical retrievals are derived for COT>4, it seems logical to use the same threshold for comparing SEVIRI with MODIS.

Maps: It is very difficult to extract quantitative information from the maps due to the use of a continuous color palette (too many tones). Instead, it would be better to define

only 10 or 12 discrete colors.

---

## Referee Comment (RC2) · Anonymous Referee #2 · 19 Jul 2018

This paper presents an inter-comparison of cloud liquid water path retrievals using several different satellite retrievals toward the goals of evaluating the retrievals resulting from the SEVIRI geostationary observations relative to other satellite sensors, characterizing the specific biases resulting from light-absorbing aerosol above the cloud, and describing the diurnal cycle of stratocumulus clouds in the southeast Atlantic Ocean region. Given that the paper tackles all three of those goals, it is perhaps a bit ambitious. However, these issues are coupled together and the paper provides a reasonably comprehensive study of the relative biases in the retrievals and the diurnal cycle. I believe the paper may be published following some relatively minor revisions to some aspects of the paper discussed further below.

[Figure]

1) Figure 2 indicates that the SEVIRI retrieval is substantially more sensitive to the presence of smoke above the cloud. The difference in the response is robust and indicates something meaningful about the differences in how the retrievals are performed, but is only addressed very briefly on line 375 as "partially explained by the spectral difference that for SEVIRI retrievals the 0.6um channel is used as a non-absorbing channel in contrast to the 0.8um channel for MODIS." I feel that this needs some deeper discussion. In what way is the SEVIRI channel more sensitive? Perhaps there is a citation that documents that spectral absorption features that explain this. Could the MODIS bands be chosen for SEVIRI in light of this additional bias due to absorbing aerosol? If the is only partly explained by the differing spectral absorption of smoke between 0.6 um and 0.8 um, then what are the other contributing factors?

2) In line 319 it is noted that the SEVIRI retrieval exhibits a strong decrease in effective radius with increasing smoke above the cloud, but only a very weak decrease in cloud optical thickness. Is this consistent the cases presented in the Haywood et al. (2004) paper? Many of the cases in that paper exhibited a strong decrease in the optical thickness and only a weak decrease in the retrieved effective radius, although the details depend on the spectral bands chosen for the retrieval. Also, is this consistent with the explanation offered above for the stronger sensitivity of SEVIRI to smoke? I would expect that if the so-called "non-absorbing" band chosen is substantially more sensitive to smoke absorption, that this would cause a more substantial impact on the retrieved optical thickness than the effective radius. This needs to be clarified.

3) In the paragraph beginning line 435 comparing SEVIRI and MODIS is broken cloud scenes, it is noted that SEVIRI is biased high relative to MODIS primarily because of a high bias in the effective radius retrieved. The authors argue that this could be caused by the SEVIRI algorithm's artificial use of a climatological effective radius for optically thin clouds. However, I wonder if it might also be contributed by the differences in resolution between SEVIRI and MODIS. Could it be that SEVIRI with a larger footprint than MODIS is simply more likely in broken cloud scenes to report a valid retrieval in a

pixel that in reality is contaminated by some inhomogeneity or clear-sky regions? That would presumably lead to a high bias in effective radius that is more substantial for SEVIRI than MODIS.

---

## Author Comment (AC1) · 16 Aug 2018

pdf file includes replies to reviewer#1 and reviewer#2, revised manuscript (track changed version), and revised supplemental material (track changed version)

Please also note the supplement to this comment:
https://www.atmos-chem-phys-discuss.net/acp-2018-445/acp-2018-445-AC1-supplement.pdf

---

## Author Response (ED1)

**Response to Anonymous Referee #1**

The authors would like to thank Anonymous Referee #1 for his comments. Below, please find our response to the referee's comments (RC to denote Reviewer's comment, AR to denote Authors' reply).

RC-1) Main objective: It is unclear if the goal of the study is to understand the seasonal/diurnal cloud evolution of the Namibia-Angola stratiform cloud regime (as suggested by the title) or to evaluate SEVIRI cloud retrievals with other datasets (most of the analysis revolves around the differences between SEVIRI and other datasets, and potential bias in SEVIRI retrievals). If the focus is to characterize the diurnal cycle, then please provide more detailed information about the amplitude of the daytime cycle and explain spatial/temporal changes in the context of the atmospheric circulation and thermodynamical structure. For instance, in Painemal et al. (2012, JGR), we attempted to understand the dynamical factors behind the cloud diurnal cycle in the SE Pacific, and showed hourly composites (maps) of cloud retrievals. In Painemal et al. (2013, J. Atmos Sc.), we further endeavored to understand variations in liquid water path and cloud fraction, in the context of the boundary layer depth evolution and subsidence variability. Similarly, we utilized a super-parameterized climate model and NASA-Langley SEVIRI retrievals for describing the diurnal evolution of cloud fraction and height over the Namibia-Angola stratocumulus cloud deck (Painemal et al., 2015 J. Climate). If the focus is mostly evaluating the ability of SEVIRI to reproduce the diurnal cycle, please modify the title and the introduction accordingly.

AR-1) We do agree with the reviewer's comment that the original title could mislead the reader about the objective of the study. Therefore, we modified the title to "Evaluating the diurnal cycle of South Atlantic stratocumulus clouds as observed by MSG-SEVIRI". The primary objective of our manuscript is to *evaluate* the diurnal cycle of South Atlantic stratocumulus clouds based on cloud property retrievals from the SEVIRI CLAAS-2 algorithm. In order to make this clearer we modified the text in introduction section in the original manuscript **in line# 118**.

RC-2) Inhomogeneity and cloud mask: I was surprised that cloud fraction (mask) differences between SEVIRI CLAAS and MODIS collection 6 were not analyzed. I would speculate that the spatial pattern of the SEVIRI-MODIS difference near the equator in Figure S5d is most likely due to cloud fraction differences between both sensors/algorithms. Also, the use of

some sort of inhomogeneity index would provide support to the hypothesis that pixel resolution is in part responsible for the COT difference between SEVIRI and MODIS. Although the plane parallel bias is likely playing a role, it is puzzling that both SEVIRI COT and reff are generally smaller than their MODIS counterparts (1.6um), as one would expect that reff (COT) is overestimated (underestimated) as the pixel resolution becomes coarser (e.g. table 1a and 1b in Painemal et al., 2012 JGR). This points to other issues associated with differences in the retrieving algorithms, since SEVIRI visible channels were calibrated against MODIS. The question could be answered if the CLAAS algorithm were applied to MODIS (I do not know if this is even possible). At the very least, the authors should speculate about the causes for the inconsistencies between MODIS and SEVIRI that cannot be explained by the plane parallel bias or absorbing aerosols. Lastly, if the COT threshold was applied to SEVIRI (and MODIS), then a comparable threshold should be applied to TMI. If not, the comparison between TMI and SEVIRI.

AR-2) Thank you for this suggestion. In the revised manuscript we included the diurnal cycle of cloud fraction for 2-yr mean in Fig. 10 and for seasonal means in Fig. S8. The cloud fraction maps for SEVIRI, MODIS and their differences are shown in Figure 7 for all-sky condition, and described in the revised manuscript. Figure S5d in the original manuscript (Fig. 6d in the revised manuscript) is SEVIRI-MODIS 1.6-micron LWP differences for the overcast case. Therefore, SEVIRI and MODIS LCF are >95 % and also both LCFs agree within ±1 %.

We performed a preliminary analysis of factors that might explain SEVIRI COT/CER low biases. Firstly, spatial heterogeneity was considered. Similar to Painemal et al. (2013b), we found both for SEVIRI and for MODIS a decrease in COT and increase in CER with increasing scene heterogeneity under constant TMI LWP. However, the negative SEVIRI-MODIS CER difference remained a robust feature, independent of the magnitude of the scene heterogeneity. Secondly, the different view geometries of SEVIRI (fixed) and MODIS (varying from orbit to orbit) were analysed. A dependence of the SEVIRI-MODIS CER bias on the MODIS view zenith angle (VZA) was observed, with the bias varying between -0.5 µm to -2 µm and generally being lowest for the MODIS oblique backscatter view direction. Further analysis in this direction, including also the relative azimuth angle, is promising but deferred to a future study, in which also spectral and algorithmic differences between the MODIS and SEVIRI instruments and cloud property retrievals should be considered. The description is embedded **in original manuscript line#586** in the revised manuscript.

SEVIRI CER being smaller than MODIS is indeed unexpected based on the plane parallel bias effect alone. Of course, there are algorithmic differences too. For example, SEVIRI uses the 0.6-micron channel instead of the 0.8-micron as MODIS does over the ocean. Redoing the full CLAAS retrievals with the SEVIRI 0.8 micron channel or applying the CLAAS algorithms to MODIS could indeed shed light on the observed differences, but this would imply a huge effort and we feel this is beyond the scope of this study (which is an evaluation of the current official CLAAS-2 cloud properties).

Regarding the COT threshold applied to SEVIRI data, we confirm that a comparable threshold was effectively applied to both MODIS and TMI data. In the comparison we only included 0.25° x 0.25° gridboxes (and the corresponding SEVIRI, TMI, and MODIS retrievals) with a gridbox-mean SEVIRI COT > 3. All gridboxes (and the corresponding SEVIRI, TMI, and MODIS retrievals) with a mean SEVIRI COT < 3 were excluded from the analysis. **We clarified this in the first paragraph of section 3 in line# 245** in the revised manuscript.

RC-3) Inconsistent results: The curve "SEV" in Fig. 2a is SEVIRI collocated with MODIS, correct? Does it mean that not always MODIS and SEVIRI are collocated? All the retrievals should be spatially and temporally collocated. I also noticed that in Fig S9, SEVIRI reff for overcast and all-sky samples is almost the same (Fig. S9a and b). In contrast, the all-sky and overcast averages are quite different for MODIS, why?

AR-3) 'SEVIRI' is SEVIRI collocated with TMI. 'SEV' is SEVIRI collocated with MODIS. Thus, yes SEVIRI observations are always collocated in time and space with either TMI or MODIS. Unfortunately collocations in time and space of SEVIRI, TMI, and MODIS, so called triple collocations, are very rare, and would only be available when TMI and MODIS line-up. Therefor we did not use triple collocations. The fact that SEVIRI CER between overcast and all-sky are very similar, while MODIS shows larger differences, could be explained by the weighting applied in CLAAS. For thin clouds (which are the ones added in the all-sky averages) CLAAS-2 will tend to retrieve lower CER because of the weighting with a climatological a priori of 8-microns. MODIS will probably tend to retrieve higher CER for thin clouds because these are often also broken clouds for which CER is overestimated (esp. with 1.6-micron). Indeed, the differences between all-sky and overcast

MODIS CER are much smaller for the 3.7 micron channel (see Fig. S7), which is known to be less affected by cloud heterogeneity.

RC-4) What is the purpose of showing the three MODIS reff's as the apples-to-apples comparison is between MODIS-SEVIRI at 1.6 um? I understand that this is useful for understanding the reff bias due to absorbing aerosols (which is not a novel result), but it is extremely confusing to understand figures 6-7 with so many symbols and retrievals, and the overall objective of including MODIS 2.1 and 3.7 reff is unclear. Similarly, just report one MODIS COT as the three MODIS COT are essentially the same. The authors mention that differences in MODIS reff at 1.6, 2.1, and 3.7 um might be providing information about the cloud vertical structure and precipitation; however, numerous papers (e.g. Zhang and Platnick, 2011, Painemal et al. 2013 ACP: "The impact of horizontal heterogeneities, cloud fraction, and liquid water path on warm cloud effective radii from CERES-like Aqua MODIS retrievals) have shown that the difference between the 3.7 um and 2.1 um reff (or 1.6 um) mostly reflect the effect of spatial inhomogeneity and clear-sky contamination in the retrievals (for stratiform clouds).

AR-4) We agree that the different MODIS channel retrievals are useful for understanding the absorbing aerosol effect. Therefore, we propose to keep them in the new Figs. 2 and 3. We agree that the different MODIS channels don't add much in the scatter density plots and may cause confusion (new Fig. 8). Therefore, we removed the 2.1 and 3.7 micron plots from Fig. 8. In the overcast maps (Fig. 6) and diurnal cycle plots we do think the different channel MODIS CER retrievals (Fig. S7) do add important information regarding heterogeneity effects (as correctly argued by the referee). So we propose to keep these in the new version, while for LWP and COT the different MODIS retrievals are not shown anymore.

RC-5) Figs. S2 and S5 deserve to be included in the manuscript:
AR-5) We have included Figures S2 and S5 as Figs. 3 and 6 in the revised manuscript.

RC-6) Remove 'Discussion' from the title of Section 4:
AR-6) 'Discussion' has been removed from the respective section title in the revised version.

RC-7) Page 12, line 349, I disagree, Figure S1 does not show any difference for COT:
AR-7) We believe it does. Fig. S1c does show a median SEVIRI-MODIS COT bias of -1.

RC-8) Page 5, first paragraph: This is non-raining pixels according to the RSS algorithm. Depending on the threshold used to define a rainy pixel, drizzle or light precipitation is still possible:

AR-8) This is indeed possible. We have replaced the word 'avoid' by 'minimize' in the revised manuscript to indicate that not all rainy pixels will be filtered out in practice.

RC-9) Page 15, line 435: I cannot find the figure that shows the overestimation of SEVIRI (relative to MODIS) for broken scenes.

AR-9) This remark in particular concerns the 3.7-micron MODIS retrievals. Fig. S5l in the original manuscript (Fig. 6l in revised version) does show SEVIRI being 2-4 micron lower than MODIS, but for the other channels, the difference is less. However, this refers to 'the area outside the Sc region' not to 'broken clouds', since all plots in Fig. 6l are for overcast clouds only. **The discussion is elaborated in the revised version in line# 435 of the original manuscript.**

RC-10) Page 16, first paragraph. This explanation is unlikely, since you removed samples with high liquid water path (precipitating samples according to the RSS algorithms). Moreover, the spectral difference in reff is mostly indicative of the effect of spatial inhomogeneity and 3D radiative effects in the retrievals rather than information about the vertical structure or precipitation.

AR-10) Agree: the reason that MODIS 3.7 Reff is higher than that of 2.1 and 1.6 is probably mainly related to inhomogeneity. Horizontal inhomogeneity has a larger impact in the 1.6-micron than in the 3.7-micron channel. A deeper discussion is included in the revised manuscript **in line# 435 of the original manuscript**.

RC-11) Page 17, line 487, what do you mean by "smaller …".

AR-11) The relative variation in effective radius is smaller than the relative variation in COT during the day. The text in the revised version is now modified for clarity.

RC-12) Page 19 lines 547-549: I suspect this is mostly due to cloud thinning. If cloud fraction (mask) played a role, then cloud effective radius would be biased high due to clear-sky contamination and 3D radiative effects.

AR-12) We now included the diurnal cycle of SEVIRI liquid cloud fraction for our Sc study domain, both 2-yr (Fig. 10) and seasonal (Fig. S8) means. The diurnal variation in cloud fraction is very similar to the variation in all-sky LWP. COT and CER are in-cloud means, whereas LWP is in-cloud water content multiplied by liquid cloud fraction in order to compare with the gridbox mean TMI at a $0.25^{o}$x$0.25^{o}$ resolution aggregated from the original SEVIRI (and MODIS) resolution. Therefore, the diurnal variation in TMI and SEVIRI LWP follow the diurnal variation of cloud fraction in all-sky scene. For the overcast case, most of the diurnal variation in LWP is likely because of cloud thinning. In the all-sky case, MODIS 1.6 and 2.1 channel CER are indeed biased high due to clear-sky contamination and cloud heterogeneity impacts. On the other hand, 3.7 micron CER agrees well with SEVIRI. SEVIRI values are lower because of the climatological weighting applied to CER corresponding to the lower COT pixels.

RC-13) Page 20 line 558: Use austral winter instead:
AR-13) Modified accordingly in the revised version.

RC-14) Page 4, line 102, replace "evaluated" with "analyzed".
AR-14) Modified accordingly in the revised manuscript.

RC-15) Page 4, line 110, Add "In contrast," before Painemal et al. (2012). Painemal et al. (2012) also utilized in-situ cloud probe to assess the bias in satellite cloud properties.
AR-15) Have been added to the revised manuscript version.

RC-16) Page 5 line 143: Define VIS/NIR.
AR-16) In the revised version this has been defined.

RC-17) Page 6, line 157, If the physical retrievals are derived for COT>4, it seems logical to use the same threshold for comparing SEVIRI with MODIS.

AR-17) For COT=4 there is still a 13% influence of the a priori CER according to Eq. (2), so to completely get rid of this influence, one would even have to apply a larger threshold (say

COT>5). For COT=3 the weight of the a priori CER is 35%. This seemed a reasonable compromise between a modest weighting with the a priori on the one hand and keeping enough samples for robust statistics on the other hand.

RC-18) Maps: It is very difficult to extract quantitative information from the maps due to the use of a continuous color palette (too many tones). Instead, it would be better to define only 10 or 12 discrete colors.

AR-18) In the revised version, all the maps are plotted with discrete color palettes. Indeed, the figures are more readable now.

**Response to Anonymous Referee #2**

The authors would like to thank Anonymous Referee #2 for his/her comments. Below, please find our response to the referee's comments (RC to denote Reviewer's comment, AR to denote Authors' reply).

RC-1) Figure 2 indicates that the SEVIRI retrieval is substantially more sensitive to the presence of smoke above the cloud. The difference in the response is robust and indicates something meaningful about the differences in how the retrievals are performed, but is only addressed very briefly on line 375 as "partially explained by the spectral difference that for SEVIRI retrievals the 0.6um channel is used as a non-absorbing channel in contrast to the 0.8um channel for MODIS." I feel that this needs some deeper discussion. In what way is the SEVIRI channel more sensitive? Perhaps there is a citation that documents that spectral absorption features that explain this. Could the MODIS bands be chosen for SEVIRI in light of this additional bias due to absorbing aerosol? If the is only partly explained by the differing spectral absorption of smoke between 0.6 um and 0.8 um, then what are the other contributing factors?

AR-1) Indeed, a study by Haywood et al. (2004) investigated the spectral dependence of aerosol optical thickness for different non-absorbing and water absorbing channels and found that the effect of aerosol on the 0.63-micron radiance is significantly larger than that on the 0.86 micron radiance. The presence of aerosols above clouds reduces the 0.63-micron radiance substantially more than the 0.86-micron radiance. This could potentially introduce a low bias of (20% to >30%) 2 to >6 in retrieved cloud optical thickness for clouds with true COT 10 and 20, depending upon how large the true COT is, as the bias is highest for clouds with the largest COT. This low bias is larger for the retrieval at 0.63-micron SEVIRI channel than at 0.86-micron MODIS non-absorbing visible channel. Depending upon the paired water-absorbing NIR channel, the CER retrieval is also biased. Combining the 3.7-micron channel with the 0.63 or 0.86-micron visible channel, there is a relatively modest CER high bias of <1 micron, as the constant CER lines are more or less parallel to the 0.63 or 0.86-micron axis. However, the radiance pair of 0.86/1.63 introduced a significant low bias in CER of about 3 micron for a cloud with actual CER of 10 micron due to the apparent indirect effect induced by the decreased 0.86-micron radiance on non-parallel constant CER lines. This low bias will be even larger for the 0.6/1.63 radiance pair as used for the CLAAS-2 SEVIRI retrievals, and hence both SEVIRI COT and CER are expected to be lower than the

corresponding MODIS values retrieved from the 0.86/1.63 micron radiance pair. **This description is embedded in line# 347 of the original manuscript.**

RC-2) In line 319 it is noted that the SEVIRI retrieval exhibits a strong decrease in effective radius with increasing smoke above the cloud, but only a very weak decrease in cloud optical thickness. Is this consistent the cases presented in the Haywood et al. (2004) paper? Many of the cases in that paper exhibited a strong decrease in the optical thickness and only a weak decrease in the retrieved effective radius, although the details depend on the spectral bands chosen for the retrieval. Also, is this consistent with the explanation offered above for the stronger sensitivity of SEVIRI to smoke? I would expect that if the so-called "non-absorbing" band chosen is substantially more sensitive to smoke absorption, that this would cause a more substantial impact on the retrieved optical thickness than the effective radius. This needs to be clarified.

AR-2) The sentence in line 319 in original manuscript, about the "only a very weak decrease in cloud optical thickness with increasing AI", is further described (in the original manuscript itself) in the paragraph immediately following it **in line# 330**. As such, cloud optical thickness has shown a weak decrease, but the expectation is that the true COT has to increase with AI, as strongly suggested by the sharp TMI LWP increase with AI. Thus, the overall low bias in COT due to smoke is indeed more substantial, in agreement with Haywood et al. (2004), because there is a simultaneous increase in true COT with AI (i.e. getting closer to the coast). The low bias in SEVIRI CER will also be larger than in MODIS CER due to the use of the 0.63/1.6-micron spectral pair, as the 0.63-micron reflectance would be substantially more affected by smoke than the 0.86-micron reflectance used in MODIS and also because the 1.6-micron based constant CER lines are less parallel to the 0.63-micron reflectance axis than to the 0.86-micron reflectance axis of the MODIS LUTs.

RC-3) In the paragraph beginning in line 435 comparing SEVIRI and MODIS in broken cloud scenes, it is noted that SEVIRI is biased high relative to MODIS primarily because of a high bias in the effective radius retrieved. The authors argue that this could be caused by the SEVIRI algorithm's artificial use of a climatological effective radius for optically thin clouds. However, I wonder if it might also be contributed by the differences in resolution between SEVIRI and MODIS. Could it be that SEVIRI with a larger footprint than MODIS is simply more likely in broken cloud scenes to report a valid retrieval in a pixel that in reality

is contaminated by some inhomogeneity or clear-sky regions? That would presumably lead to a high bias in effective radius that is more substantial for SEVIRI than MODIS.

AR-3) The results presented in line 435 (in original manuscript) are for overcast gridboxes, where overcast is selected with the criteria of LCF>95% and COT>3. Outside the identified Sc regime, near the equator it is possible that the positive plane-parallel CER bias in heterogeneous cloud scenes with low COTs is the dominant one for the larger SEVIRI pixel size, whereas towards south it seems that the effect of climatological weighting applied to SEVIRI CER is dominant. The MODIS 1.6 and 2.1 micron CERs could have been impacted by plane-parallel bias and overestimated the retrieved CER. **This description is embedded in line# 428 in the original manuscript.**

[revised manuscript text omitted]
 CER underestimation is not. **Can you offer some possible explanation? Is it related to the algorithms or is there some physical explanation?**

    We performed a preliminary analysis of factors that might explain these biases. Firstly, spatial heterogeneity was considered. Similar to Painemal et al. (2013b), we found both for SEVIRI and for MODIS a decrease in COT and increase in CER with increasing scene heterogeneity under constant TMI LWP. However, the

810 negative SEVIRI-MODIS CER difference remained a robust feature, independent of the magnitude of the scene heterogeneity. Secondly, the different view geometries of SEVIRI (fixed) and MODIS (varying from orbit to orbit) were analysed. A dependence of the SEVIRI-MODIS CER bias on the MODIS view zenith angle (VZA) was observed, with the bias varying between -0.5 μm to -2 μm and generally being lowest for the MODIS oblique backscatter view direction. Further analysis in this direction, including also the relative azimuth angle, is promising

815 but deferred to a future study, in which also spectral and algorithmic differences between the MODIS and SEVIRI instruments and cloud property retrievals should be considered. **Cumbersome sentence. Breakup and re-phrase.**

[revised manuscript text omitted]

---

## Author Response (AR2)

Dear Co-Editor,

We thank you for accepting our manuscript (acp-2018-445) for publication in ACP. The recommended technical corrections have been implemented in the revised manuscript (see the attached track-changed version). COT, representing cloud optical thickness, and CER, representing cloud droplet effective radius, have respectively been replaced by '$\tau$' and '$r_e$' throughout the manuscript. The liquid water path (LWP) computation in Eq 1, line#154 is performed assuming plane-parallel and vertically homogeneous clouds, which is clarified in the revised manuscript. In addition, the potential reasons for the SEVIRI low $r_e$ bias compared to MODIS $r_e$ are further elaborated and clarified in the 'Summary' section of the revised manuscript.

Sincerely,
Seethala Chellappan and Co-authors

**Response to Anonymous Referee #1**

The authors would like to thank Anonymous Referee #1 for his comments. Below, please find our response to the referee's comments (RC to denote Reviewer's comment, AR to denote Authors' reply).

RC-1) Main objective: It is unclear if the goal of the study is to understand the seasonal/diurnal cloud evolution of the Namibia-Angola stratiform cloud regime (as suggested by the title) or to evaluate SEVIRI cloud retrievals with other datasets (most of the analysis revolves around the differences between SEVIRI and other datasets, and potential bias in SEVIRI retrievals). If the focus is to characterize the diurnal cycle, then please provide more detailed information about the amplitude of the daytime cycle and explain spatial/temporal changes in the context of the atmospheric circulation and thermodynamical structure. For instance, in Painemal et al. (2012, JGR), we attempted to understand the dynamical factors behind the cloud diurnal cycle in the SE Pacific, and showed hourly composites (maps) of cloud retrievals. In Painemal et al. (2013, J. Atmos Sc.), we further endeavored to understand variations in liquid water path and cloud fraction, in the context of the boundary layer depth evolution and subsidence variability. Similarly, we utilized a super-parameterized climate model and NASA-Langley SEVIRI retrievals for describing the diurnal evolution of cloud fraction and height over the Namibia-Angola stratocumulus cloud deck (Painemal et al., 2015 J. Climate). If the focus is mostly evaluating the ability of SEVIRI to reproduce the diurnal cycle, please modify the title and the introduction accordingly.

AR-1) We do agree with the reviewer's comment that the original title could mislead the reader about the objective of the study. Therefore, we modified the title to "Evaluating the diurnal cycle of South Atlantic stratocumulus clouds as observed by MSG-SEVIRI". The primary objective of our manuscript is to *evaluate* the diurnal cycle of South Atlantic stratocumulus clouds based on cloud property retrievals from the SEVIRI CLAAS-2 algorithm. In order to make this clearer we modified the text in introduction section in the original manuscript **in line# 118**.

RC-2) Inhomogeneity and cloud mask: I was surprised that cloud fraction (mask) differences between SEVIRI CLAAS and MODIS collection 6 were not analyzed. I would speculate that the spatial pattern of the SEVIRI-MODIS difference near the equator in Figure S5d is most likely due to cloud fraction differences between both sensors/algorithms. Also, the use of

some sort of inhomogeneity index would provide support to the hypothesis that pixel resolution is in part responsible for the COT difference between SEVIRI and MODIS. Although the plane parallel bias is likely playing a role, it is puzzling that both SEVIRI COT and reff are generally smaller than their MODIS counterparts (1.6um), as one would expect that reff (COT) is overestimated (underestimated) as the pixel resolution becomes coarser (e.g. table 1a and 1b in Painemal et al., 2012 JGR). This points to other issues associated with differences in the retrieving algorithms, since SEVIRI visible channels were calibrated against MODIS. The question could be answered if the CLAAS algorithm were applied to MODIS (I do not know if this is even possible). At the very least, the authors should speculate about the causes for the inconsistencies between MODIS and SEVIRI that cannot be explained by the plane parallel bias or absorbing aerosols. Lastly, if the COT threshold was applied to SEVIRI (and MODIS), then a comparable threshold should be applied to TMI. If not, the comparison between TMI and SEVIRI.

AR-2) Thank you for this suggestion. In the revised manuscript we included the diurnal cycle of cloud fraction for 2-yr mean in Fig. 10 and for seasonal means in Fig. S8. The cloud fraction maps for SEVIRI, MODIS and their differences are shown in Figure 7 for all-sky condition, and described in the revised manuscript. Figure S5d in the original manuscript (Fig. 6d in the revised manuscript) is SEVIRI-MODIS 1.6-micron LWP differences for the overcast case. Therefore, SEVIRI and MODIS LCF are >95 % and also both LCFs agree within ±1 %.

We performed a preliminary analysis of factors that might explain SEVIRI COT/CER low biases. Firstly, spatial heterogeneity was considered. Similar to Painemal et al. (2013b), we found both for SEVIRI and for MODIS a decrease in COT and increase in CER with increasing scene heterogeneity under constant TMI LWP. However, the negative SEVIRI-MODIS CER difference remained a robust feature, independent of the magnitude of the scene heterogeneity. Secondly, the different view geometries of SEVIRI (fixed) and MODIS (varying from orbit to orbit) were analysed. A dependence of the SEVIRI-MODIS CER bias on the MODIS view zenith angle (VZA) was observed, with the bias varying between -0.5 μm to -2 μm and generally being lowest for the MODIS oblique backscatter view direction. Further analysis in this direction, including also the relative azimuth angle, is promising but deferred to a future study, in which also spectral and algorithmic differences between the MODIS and SEVIRI instruments and cloud property retrievals should be considered. The description is embedded **in original manuscript line#586** in the revised manuscript.

SEVIRI CER being smaller than MODIS is indeed unexpected based on the plane parallel bias effect alone. Of course, there are algorithmic differences too. For example, SEVIRI uses the 0.6-micron channel instead of the 0.8-micron as MODIS does over the ocean. Redoing the full CLAAS retrievals with the SEVIRI 0.8 micron channel or applying the CLAAS algorithms to MODIS could indeed shed light on the observed differences, but this would imply a huge effort and we feel this is beyond the scope of this study (which is an evaluation of the current official CLAAS-2 cloud properties).

Regarding the COT threshold applied to SEVIRI data, we confirm that a comparable threshold was effectively applied to both MODIS and TMI data. In the comparison we only included 0.25° x 0.25° gridboxes (and the corresponding SEVIRI, TMI, and MODIS retrievals) with a gridbox-mean SEVIRI COT > 3. All gridboxes (and the corresponding SEVIRI, TMI, and MODIS retrievals) with a mean SEVIRI COT < 3 were excluded from the analysis. **We clarified this in the first paragraph of section 3 in line# 245** in the revised manuscript**.**

RC-3) Inconsistent results: The curve "SEV" in Fig. 2a is SEVIRI collocated with MODIS, correct? Does it mean that not always MODIS and SEVIRI are collocated? All the retrievals should be spatially and temporally collocated. I also noticed that in Fig S9, SEVIRI reff for overcast and all-sky samples is almost the same (Fig. S9a and b). In contrast, the all-sky and overcast averages are quite different for MODIS, why?

AR-3) 'SEVIRI' is SEVIRI collocated with TMI. 'SEV' is SEVIRI collocated with MODIS. Thus, yes SEVIRI observations are always collocated in time and space with either TMI or MODIS. Unfortunately collocations in time and space of SEVIRI, TMI, and MODIS, so called triple collocations, are very rare, and would only be available when TMI and MODIS line-up. Therefor we did not use triple collocations. The fact that SEVIRI CER between overcast and all-sky are very similar, while MODIS shows larger differences, could be explained by the weighting applied in CLAAS. For thin clouds (which are the ones added in the all-sky averages) CLAAS-2 will tend to retrieve lower CER because of the weighting with a climatological a priori of 8-microns. MODIS will probably tend to retrieve higher CER for thin clouds because these are often also broken clouds for which CER is overestimated (esp. with 1.6-micron). Indeed, the differences between all-sky and overcast

MODIS CER are much smaller for the 3.7 micron channel (see Fig. S7), which is known to be less affected by cloud heterogeneity.

RC-4) What is the purpose of showing the three MODIS reff's as the apples-to-apples comparison is between MODIS-SEVIRI at 1.6 um? I understand that this is useful for understanding the reff bias due to absorbing aerosols (which is not a novel result), but it is extremely confusing to understand figures 6-7 with so many symbols and retrievals, and the overall objective of including MODIS 2.1 and 3.7 reff is unclear. Similarly, just report one MODIS COT as the three MODIS COT are essentially the same. The authors mention that differences in MODIS reff at 1.6, 2.1, and 3.7 um might be providing information about the cloud vertical structure and precipitation; however, numerous papers (e.g. Zhang and Platnick, 2011, Painemal et al. 2013 ACP: "The impact of horizontal heterogeneities, cloud fraction, and liquid water path on warm cloud effective radii from CERES-like Aqua MODIS retrievals) have shown that the difference between the 3.7 um and 2.1 um reff (or 1.6 um) mostly reflect the effect of spatial inhomogeneity and clear-sky contamination in the retrievals (for stratiform clouds).

AR-4) We agree that the different MODIS channel retrievals are useful for understanding the absorbing aerosol effect. Therefore, we propose to keep them in the new Figs. 2 and 3. We agree that the different MODIS channels don't add much in the scatter density plots and may cause confusion (new Fig. 8). Therefore, we removed the 2.1 and 3.7 micron plots from Fig. 8. In the overcast maps (Fig. 6) and diurnal cycle plots we do think the different channel MODIS CER retrievals (Fig. S7) do add important information regarding heterogeneity effects (as correctly argued by the referee). So we propose to keep these in the new version, while for LWP and COT the different MODIS retrievals are not shown anymore.

RC-5) Figs. S2 and S5 deserve to be included in the manuscript:
AR-5) We have included Figures S2 and S5 as Figs. 3 and 6 in the revised manuscript.

RC-6) Remove 'Discussion' from the title of Section 4:
AR-6) 'Discussion' has been removed from the respective section title in the revised version.

RC-7) Page 12, line 349, I disagree, Figure S1 does not show any difference for COT:
AR-7) We believe it does. Fig. S1c does show a median SEVIRI-MODIS COT bias of -1.

RC-8) Page 5, first paragraph: This is non-raining pixels according to the RSS algorithm. Depending on the threshold used to define a rainy pixel, drizzle or light precipitation is still possible:

AR-8) This is indeed possible. We have replaced the word 'avoid' by 'minimize' in the revised manuscript to indicate that not all rainy pixels will be filtered out in practice.

RC-9) Page 15, line 435: I cannot find the figure that shows the overestimation of SEVIRI (relative to MODIS) for broken scenes.

AR-9) This remark in particular concerns the 3.7-micron MODIS retrievals. Fig. S5l in the original manuscript (Fig. 6l in revised version) does show SEVIRI being 2-4 micron lower than MODIS, but for the other channels, the difference is less. However, this refers to 'the area outside the Sc region' not to 'broken clouds', since all plots in Fig. 6l are for overcast clouds only. **The discussion is elaborated in the revised version in line# 435 of the original manuscript.**

RC-10) Page 16, first paragraph. This explanation is unlikely, since you removed samples with high liquid water path (precipitating samples according to the RSS algorithms). Moreover, the spectral difference in reff is mostly indicative of the effect of spatial inhomogeneity and 3D radiative effects in the retrievals rather than information about the vertical structure or precipitation.

AR-10) Agree: the reason that MODIS 3.7 Reff is higher than that of 2.1 and 1.6 is probably mainly related to inhomogeneity. Horizontal inhomogeneity has a larger impact in the 1.6-micron than in the 3.7-micron channel. A deeper discussion is included in the revised manuscript **in line# 435 of the original manuscript**.

RC-11) Page 17, line 487, what do you mean by "smaller …".

AR-11) The relative variation in effective radius is smaller than the relative variation in COT during the day. The text in the revised version is now modified for clarity.

RC-12) Page 19 lines 547-549: I suspect this is mostly due to cloud thinning. If cloud fraction (mask) played a role, then cloud effective radius would be biased high due to clear-sky contamination and 3D radiative effects.

AR-12) We now included the diurnal cycle of SEVIRI liquid cloud fraction for our Sc study domain, both 2-yr (Fig. 10) and seasonal (Fig. S8) means. The diurnal variation in cloud fraction is very similar to the variation in all-sky LWP. COT and CER are in-cloud means, whereas LWP is in-cloud water content multiplied by liquid cloud fraction in order to compare with the gridbox mean TMI at a $0.25^o \times 0.25^o$ resolution aggregated from the original SEVIRI (and MODIS) resolution. Therefore, the diurnal variation in TMI and SEVIRI LWP follow the diurnal variation of cloud fraction in all-sky scene. For the overcast case, most of the diurnal variation in LWP is likely because of cloud thinning. In the all-sky case, MODIS 1.6 and 2.1 channel CER are indeed biased high due to clear-sky contamination and cloud heterogeneity impacts. On the other hand, 3.7 micron CER agrees well with SEVIRI. SEVIRI values are lower because of the climatological weighting applied to CER corresponding to the lower COT pixels.

RC-13) Page 20 line 558: Use austral winter instead:
AR-13) Modified accordingly in the revised version.

RC-14) Page 4, line 102, replace "evaluated" with "analyzed".
AR-14) Modified accordingly in the revised manuscript.

RC-15) Page 4, line 110, Add "In contrast," before Painemal et al. (2012). Painemal et al. (2012) also utilized in-situ cloud probe to assess the bias in satellite cloud properties.
AR-15) Have been added to the revised manuscript version.

RC-16) Page 5 line 143: Define VIS/NIR.
AR-16) In the revised version this has been defined.

RC-17) Page 6, line 157, If the physical retrievals are derived for COT>4, it seems logical to use the same threshold for comparing SEVIRI with MODIS.

AR-17) For COT=4 there is still a 13% influence of the a priori CER according to Eq. (2), so to completely get rid of this influence, one would even have to apply a larger threshold (say

COT>5). For COT=3 the weight of the a priori CER is 35%. This seemed a reasonable compromise between a modest weighting with the a priori on the one hand and keeping enough samples for robust statistics on the other hand.

RC-18) Maps: It is very difficult to extract quantitative information from the maps due to the use of a continuous color palette (too many tones). Instead, it would be better to define only 10 or 12 discrete colors.

AR-18) In the revised version, all the maps are plotted with discrete color palettes. Indeed, the figures are more readable now.

**Response to Anonymous Referee #2**

The authors would like to thank Anonymous Referee #2 for his/her comments. Below, please find our response to the referee's comments (RC to denote Reviewer's comment, AR to denote Authors' reply).

RC-1) Figure 2 indicates that the SEVIRI retrieval is substantially more sensitive to the presence of smoke above the cloud. The difference in the response is robust and indicates something meaningful about the differences in how the retrievals are performed, but is only addressed very briefly on line 375 as "partially explained by the spectral difference that for SEVIRI retrievals the 0.6um channel is used as a non-absorbing channel in contrast to the 0.8um channel for MODIS." I feel that this needs some deeper discussion. In what way is the SEVIRI channel more sensitive? Perhaps there is a citation that documents that spectral absorption features that explain this. Could the MODIS bands be chosen for SEVIRI in light of this additional bias due to absorbing aerosol? If the is only partly explained by the differing spectral absorption of smoke between 0.6 um and 0.8 um, then what are the other contributing factors?

AR-1) Indeed, a study by Haywood et al. (2004) investigated the spectral dependence of aerosol optical thickness for different non-absorbing and water absorbing channels and found that the effect of aerosol on the 0.63-micron radiance is significantly larger than that on the 0.86 micron radiance. The presence of aerosols above clouds reduces the 0.63-micron radiance substantially more than the 0.86-micron radiance. This could potentially introduce a low bias of (20% to >30%) 2 to >6 in retrieved cloud optical thickness for clouds with true COT 10 and 20, depending upon how large the true COT is, as the bias is highest for clouds with the largest COT. This low bias is larger for the retrieval at 0.63-micron SEVIRI channel than at 0.86-micron MODIS non-absorbing visible channel. Depending upon the paired water-absorbing NIR channel, the CER retrieval is also biased. Combining the 3.7-micron channel with the 0.63 or 0.86-micron visible channel, there is a relatively modest CER high bias of <1 micron, as the constant CER lines are more or less parallel to the 0.63 or 0.86-micron axis. However, the radiance pair of 0.86/1.63 introduced a significant low bias in CER of about 3 micron for a cloud with actual CER of 10 micron due to the apparent indirect effect induced by the decreased 0.86-micron radiance on non-parallel constant CER lines. This low bias will be even larger for the 0.6/1.63 radiance pair as used for the CLAAS-2 SEVIRI retrievals, and hence both SEVIRI COT and CER are expected to be lower than the

corresponding MODIS values retrieved from the 0.86/1.63 micron radiance pair. **This description is embedded in line# 347 of the original manuscript.**

RC-2) In line 319 it is noted that the SEVIRI retrieval exhibits a strong decrease in effective radius with increasing smoke above the cloud, but only a very weak decrease in cloud optical thickness. Is this consistent the cases presented in the Haywood et al. (2004) paper? Many of the cases in that paper exhibited a strong decrease in the optical thickness and only a weak decrease in the retrieved effective radius, although the details depend on the spectral bands chosen for the retrieval. Also, is this consistent with the explanation offered above for the stronger sensitivity of SEVIRI to smoke? I would expect that if the so-called "non-absorbing" band chosen is substantially more sensitive to smoke absorption, that this would cause a more substantial impact on the retrieved optical thickness than the effective radius. This needs to be clarified.

AR-2) The sentence in line 319 in original manuscript, about the "only a very weak decrease in cloud optical thickness with increasing AI", is further described (in the original manuscript itself) in the paragraph immediately following it **in line# 330**. As such, cloud optical thickness has shown a weak decrease, but the expectation is that the true COT has to increase with AI, as strongly suggested by the sharp TMI LWP increase with AI. Thus, the overall low bias in COT due to smoke is indeed more substantial, in agreement with Haywood et al. (2004), because there is a simultaneous increase in true COT with AI (i.e. getting closer to the coast). The low bias in SEVIRI CER will also be larger than in MODIS CER due to the use of the 0.63/1.6-micron spectral pair, as the 0.63-micron reflectance would be substantially more affected by smoke than the 0.86-micron reflectance used in MODIS and also because the 1.6-micron based constant CER lines are less parallel to the 0.63-micron reflectance axis than to the 0.86-micron reflectance axis of the MODIS LUTs.

RC-3) In the paragraph beginning in line 435 comparing SEVIRI and MODIS in broken cloud scenes, it is noted that SEVIRI is biased high relative to MODIS primarily because of a high bias in the effective radius retrieved. The authors argue that this could be caused by the SEVIRI algorithm's artificial use of a climatological effective radius for optically thin clouds. However, I wonder if it might also be contributed by the differences in resolution between SEVIRI and MODIS. Could it be that SEVIRI with a larger footprint than MODIS is simply more likely in broken cloud scenes to report a valid retrieval in a pixel that in reality

is contaminated by some inhomogeneity or clear-sky regions? That would presumably lead to a high bias in effective radius that is more substantial for SEVIRI than MODIS.

AR-3) The results presented in line 435 (in original manuscript) are for overcast gridboxes, where overcast is selected with the criteria of LCF>95% and COT>3. Outside the identified Sc regime, near the equator it is possible that the positive plane-parallel CER bias in heterogeneous cloud scenes with low COTs is the dominant one for the larger SEVIRI pixel size, whereas towards south it seems that the effect of climatological weighting applied to SEVIRI CER is dominant. The MODIS 1.6 and 2.1 micron CERs could have been impacted by plane-parallel bias and overestimated the retrieved CER. **This description is embedded in line# 428 in the original manuscript.**

[revised manuscript text omitted]

Authors 8/28/2018 9:23 PM

Authors 8/28/2018 9:23 PM

**Supplemental Materials**

**Table S1.** Two-year mean and seasonal statistics of collocated SEVIRI and MODIS retrievals in rain-free, ice-free, smoke-free ($AI < 0.1$), $\tau > 3$, and overcast (LCF $\geq$ 95%) grid cells over the marine stratocumulus region. The $\tau$ means and $r_e$ means (in micron) are listed. The values in brackets are statistics without filtering for LCF $\geq$ 95% and $\tau > 3$, i.e., for the all-sky case.

|  | JJA | SON | DJF | MAM | Two-year |
|---|---|---|---|---|---|
| **Stratocumulus (SEVIRI vs. MODIS)** | | | | | |
| SEVIRI $\tau$ | 10.2 (5.9) | 10.3 (7.1) | 9.9 (5.2) | 9.9 (4.8) | 10.2 (6.0) |
| MODIS 1.6 $\tau$ | 11.3 (7.3) | 11.3 (8.3) | 10.8 (6.3) | 10.8 (5.9) | 11.1 (7.2) |
| SEVIRI $r_e$ | 8.8 (10.2) | 10.2 (11.1) | 11.6 (11.4) | 10.5 (10.7) | 10.1 (10.9) |
| MODIS 1.6 $r_e$ | 10.3 (12.6) | 11.5 (12.9) | 12.2 (13.4) | 11.4 (13.2) | 11.3 (13.0) |
| MODIS 2.1 $r_e$ | 11.1 (13.8) | 11.9 (13.3) | 12.3 (13.4) | 11.6 (13.6) | 11.7 (13.5) |
| MODIS 3.7 $r_e$ | 11.5 (12.6) | 11.7 (12.5) | 11.6 (11.9) | 11.2 (11.8) | 11.6 (12.2) |
| Correl. $\tau$ 1.6 | 0.96 (0.93) | 0.97 (0.96) | 0.96 (0.95) | 0.96 (0.94) | 0.96 (0.95) |
| Correl. $r_e$ 1.6 | 0.93 (0.73) | 0.90 (0.76) | 0.92 (0.60) | 0.92 (0.62) | 0.92 (0.70) |
| Correl. $r_e$ 2.1 | 0.90 (0.77) | 0.90 (0.77) | 0.91 (0.62) | 0.92 (0.67) | 0.89 (0.72) |
| Correl. $r_e$ 3.7 | 0.86 (0.81) | 0.78 (0.76) | 0.87 (0.64) | 0.87 (0.64) | 0.80 (0.74) |

**Discussion:** Frequency histograms of SEVIRI – MODIS LWP, $\tau$, and $r_e$ difference, as well as, the differences relative to MODIS LWP, $\tau$, and $r_e$ for the overcast condition aggregated during JAS 2011 and JAS 2012 are shown in Fig. S1. The histogram of SEVIRI – MODIS $\tau$ differences revealed that the peak of the distribution is off zero with ~35 % of the data falling into the -1 bin. Only ~17 % of data showed mean zero difference, while ~23 % of data showed a difference of -2. The SEVIRI $\tau$ relative to MODIS $\tau$ was within 10 % for 36 % of the data, within 20 % for 80 % of the data, and within 30 % for 95 % of the retrievals. Overall, SEVIRI $\tau$ appeared to be low by ~1 compared to MODIS $\tau$.

SEVIRI $r_e$ retrieved in the 1.6-μm channel was compared with MODIS $r_e$ values retrieved in three water absorbing channels at 1.6-, 2.1-, and 3.7-μm. Compared to the 1.6-μm MODIS $r_e$, ~70 % of SEVIRI $r_e$ have a mean difference of -1.5 μm. Compared to the 2.1- and 3.7-μm MODIS $r_e$, the difference histograms indicate larger

Authors 8/28/2018 9:29 PM
Authors 8/28/2018 9:29 PM
Formatted [2]
Authors 8/28/2018 9:29 PM
Formatted Table [4]
Authors 8/28/2018 9:29 PM
Formatted [5]
Authors 8/28/2018 9:29 PM
Formatted [6]
Authors 8/28/2018 9:29 PM
Formatted [3]
Authors 8/28/2018 9:29 PM
Formatted [7]
Authors 8/28/2018 9:29 PM
Formatted [8]
Authors 8/28/2018 9:29 PM
Formatted [9]
Authors 8/28/2018 9:29 PM
Inserted Cells [10]
Authors 8/28/2018 9:29 PM
Authors 8/28/2018 9:29 PM
Authors 8/28/2018 9:29 PM
Authors 8/28/2018 9:29 PM
Authors 8/28/2018 9:29 PM
Authors 8/28/2018 9:29 PM
Authors 8/28/2018 9:29 PM
Authors 8/28/2018 9:29 PM
Authors 8/28/2018 9:29 PM
Authors 8/28/2018 9:29 PM
Authors 8/28/2018 9:29 PM
Authors 8/28/2018 9:29 PM
Authors 8/28/2018 9:29 PM
Authors 8/28/2018 9:29 PM
Authors 8/28/2018 9:29 PM
Authors 8/28/2018 9:29 PM
Authors 8/28/2018 9:29 PM
Authors 8/28/2018 9:29 PM
Authors 8/28/2018 9:29 PM
Authors 8/28/2018 9:29 PM
Authors 8/28/2018 9:29 PM
[21]
Authors 8/28/2018 9:29 PM
Authors 8/28/2018 9:29 PM
Authors 8/28/2018 9:29 PM
Authors 8/28/2018 9:29 PM
Authors 8/28/2018 9:29 PM
Authors 8/28/2018 9:29 PM

differences: ~55 % and ~50 % of SEVIRI $r_e$ have a difference of -2.5 µm, respectively. Although SEVIRI $r_e$ are biased low compared to all three MODIS $r_e$, the ~1 µm additional low bias relative to the 2.1- and 3.7-µm $r_e$ likely indicates much smaller smoke-induced retrieval artifacts in these two channels. In general, the $r_e$ retrievals from SEVIRI tend to be lower than corresponding retrievals from the three MODIS channels, with SEVIRI having about 1.5 µm to 2.5 µm lower $r_e$ values.

The SEVIRI minus MODIS LWP distributions peak at about -10 g m$^{-2}$ irrespective of the MODIS channel used for the retrieval. The differences between MODIS 1.6-µm and SEVIRI retrievals are within 10 % for about 30 % of SEVIRI pixels, within 20 % for about 60 % of the SEVIRI pixels, and within 30 % for about 80 % of the SEVIRI pixels. However, differences between SEVIRI and MODIS 2.1-µm and 3.7-µm channel retrievals are larger, with relative differences being smaller than 10 % for about 22 % of the SEVIRI pixels against MODIS 2.1-µm and for about 16 % of the SEVIRI pixels against MODIS 3.7-µm values.

The frequency histograms of SEVIRI – MODIS LWP, $\tau$ and $r_e$ differences, as well as the difference with respect to different MODIS channels for the 2-year aggregate are shown in Fig. S3 (all-sky case) and Fig. S4 (overcast case). The peak of the LWP absolute/relative difference distribution is centred on zero, although the distribution is negatively skewed. Interestingly, in the all-sky case ~40 % of the data have shown negligible difference (zero LWP bias bin), whereas, only about 30 % of the data have shown a negligible difference in the overcast case. About 20–30 % of the data have fallen into the LWP difference bin of -10 g m$^{-2}$ in either cases. In the overcast case, ~40 % of the data have shown a relative LWP difference < 10 % and ~90 % of the data have shown a relative LWP difference < 30 %; however, for the all-sky case, only about 25 % and 60 % of the data have shown relative LWP differences < 10 % and < 30 %. Respectively, about 48 %, 84 %, 95 % of the observations show relative $\tau$ differences within 10 %, 20 %, and 30 % in the overcast case. Similarly, about 90 % of the observations show relative $r_e$ differences within 30 % in the overcast case. Histograms of both $\tau$ and $r_e$ differences reveal that the distribution is off centered. Histograms of $\tau$ differences reveal a narrow distribution which peaks at -1 especially in the overcast case; however in the all-sky case a broader peak is noticed between -1 and 0. Histograms of $r_e$ differences reveal wider distributions (especially when compared against the 2.1- and 3.7-µm channels), which peak at -1 µm in the overcast case; however, in the all-sky case a broader peak is noticed between -2 µm and -1 µm.

[Figure]

Authors 8/28/2018 9:29 PM

Authors 8/28/2018 9:29 PM

Authors 8/28/2018 9:29 PM

[Figure]

235

**Figure S1**. Histogram of SEVIRI – MODIS liquid water path differences (a), cloud optical thickness differences (c), and droplet effective radius differences (e), as well as, histogram of SEVIRI – MODIS LWP, $\tau$, $r_e$ differences relative to MODIS LWP (b), $\tau$ (d), and $r_e$ (f) for JAS 2011 and JAS 2012 for overcast (LCF ≥ 95 % and $\tau$ > 3) rain- and ice-free conditions.

Authors 8/28/2018 9:29 PM

Authors 8/28/2018 9:29 PM

Authors 8/28/2018 9:29 PM

Authors 8/28/2018 9:29 PM

Authors 8/28/2018 9:29 PM

Authors 8/28/2018 9:29 PM

Authors 8/28/2018 9:29 PM

[Figure]

**Figure S2**. Two-year mean map of (a) SEVIRI minus TMI LWP difference, (b) SEVIRI LWP, (c) TMI LWP, (d) SEVIRI τ, (e) SEVIRI 1.6-μm $r_e$, for the all-sky case. The solid black contour denotes the identified Sc region. Rain-, ice-, and aerosol-free conditions were applied.

[Figure]

Authors 8/28/2018 9:29 PM

[Figure]

265

**Figure S3**. Histogram of SEVIRI – MODIS liquid water path differences (a), cloud optical thickness differences (c), and droplet effective radius differences (e), as well as, histogram of SEVIRI – MODIS LWP, τ, $r_e$ differences relative to MODIS LWP (b), τ (d), and $r_e$ (f) for December 2010 to November 2012 for the all-sky case with rain- and ice-free conditions.

[Figure]

**Figure S4**. Histogram of SEVIRI – MODIS liquid water path differences (a), cloud optical thickness differences (c), and droplet effective radius differences (e), as well as, histogram of SEVIRI – MODIS LWP, $\tau$, $r_e$ differences relative to MODIS LWP (b), $\tau$ (d), and $r_e$ (f) for December 2010 to November 2012 for the overcast case (LCF ≥ 95 % and $\tau$ > 3) in rain- and ice-free conditions.

[Figure]

Authors 8/28/2018 9:29 PM

Authors 8/28/2018 9:29 PM

Unknown

Authors 8/28/2018 9:29 PM
**Moved down [9]:** Figure

Authors 8/28/2018 9:29 PM
**Deleted: S6**. Histogram of SEVIRI – MODIS liquid water path differences (a), cloud optical thickness differences (c), and droplet effective radius differences (e), as well as, histogram of SEVIRI – MODIS LWP, COT, CER differences relative to MODIS LWP (b), COT (d), and CER (f) for December 2010 to November 2012 for the overcast case (LCF ≥ 95% and COT > 3) in rain- and ice-free conditions. - ... [46]

Unknown

Authors 8/28/2018 9:29 PM

**Figure S5.** Seasonal mean diurnal cycle of SEVIRI LWP bias compared to TMI as well as Terra and Aqua MODIS, over the Sc region, both for all-sky and overcast-cases (LCF ≥ 95 % and $\tau$ > 3): (a) DJF, (b) MAM, (c) JJA, and (d) SON of the study period. Rain-, ice-, and smoke-free conditions were applied.

315

[Figure]

[Figure]

Authors 8/28/2018 9:29 PM

Unknown

Unknown

320

**Figure S6.** Seasonal mean diurnal cycle of SEVIRI and Terra and Aqua MODIS cloud optical thicknesses over the Sc region, both for all-sky and overcast-cases (LCF ≥ 95 % and $\tau$ > 3): (a) DJF, (b) MAM, (c) JJA, and (d) SON of the study period.

Authors 8/28/2018 9:29 PM

Authors 8/28/2018 9:29 PM

325

330

335

**Figure S7.** Seasonal mean diurnal cycle of SEVIRI and Terra and Aqua MODIS cloud droplet effective radius over the Sc region, both for all-sky and overcast-cases (LCF ≥ 95 % and τ > 3): (a) DJF, (b) MAM, (c) JJA, and (d) SON of the study period.

340

Authors 8/28/2018 9:29 PM

Authors 8/28/2018 9:29 PM

Authors 8/28/2018 9:29 PM

Authors 8/28/2018 9:29 PM

Authors 8/28/2018 9:29 PM

Authors 8/28/2018 9:29 PM

Authors 8/28/2018 9:29 PM

Authors 8/28/2018 9:29 PM

345

Authors 8/28/2018 9:29 PM
**Moved (insertion) [9]**
Authors 8/28/2018 9:29 PM

350

355 **Figure S8.** Seasonal mean diurnal cycle of SEVIRI and Terra and Aqua MODIS liquid cloud fraction over the Sc region, for all-sky case: (a) DJF, (b) MAM, (c) JJA, and (d) SON of the study period.